# Chronic muscle weakness and mitochondrial dysfunction in the absence of sustained atrophy in a preclinical sepsis model

Allison M Owen[1,2,3]\*, Samir P Patel[2,4], Jeffrey D Smith[5,6], Beverly K Balasuriya[1,3], Stephanie F Mori[1,3], Gregory S Hawk[7], Arnold J Stromberg[7], Naohide Kuriyama[8], Masao Kaneki[8], Alexander G Rabchevsky[2,4], Timothy A Butterfield[2,6], Karyn A Esser[2,6,9], Charlotte A Peterson[2,6,10], Marlene E Starr[1,3,11], Hiroshi Saito[1,2,3,12]\*

[1]Aging and Critical Care Research Laboratory, University of Kentucky, Lexington, United States; [2]Department of Physiology, University of Kentucky, Lexington, United States; [3]Department of Surgery, University of Kentucky, Lexington, United States; [4]Spinal Cord and Brain Injury Research Center, University of Kentucky, Lexington, United States; [5]Department of Biosystems and Agricultural Engineering, University of Kentucky, Lexington, United States; [6]Center for Muscle Biology, University of Kentucky, Lexington, United States; [7]Department of Statistics, University of Kentucky, Lexington, United States; [8]Department of Anesthesia, Critical Care and Pain Medicine, Massachusetts General Hospital, Shriners Hospitals for Children, Harvard Medical School, Charlestown, United States; [9]Department of Physiology and Functional Genomics, University of Florida, Gainesville, United States; [10]Department of Rehabilitation Sciences, University of Kentucky, Lexington, United States; [11]Department of Pharmacology and Nutritional Sciences, University of Kentucky, Lexington, United States; [12]Markey Cancer Center, University of Kentucky, Lexington, United States

\*For correspondence:
a.steele@uky.edu (AMO);
hiroshi.saito@uky.edu (HS)

**Competing interests:** The authors declare that no competing interests exist.

**Abstract** Chronic critical illness is a global clinical issue affecting millions of sepsis survivors annually. Survivors report chronic skeletal muscle weakness and development of new functional limitations that persist for years. To delineate mechanisms of sepsis-induced chronic weakness, we first surpassed a critical barrier by establishing a murine model of sepsis with ICU-like interventions that allows for the study of survivors. We show that sepsis survivors have profound weakness for at least 1 month, even after recovery of muscle mass. Abnormal mitochondrial ultrastructure, impaired respiration and electron transport chain activities, and persistent protein oxidative damage were evident in the muscle of survivors. Our data suggest that sustained mitochondrial dysfunction, rather than atrophy alone, underlies chronic sepsis-induced muscle weakness. This study emphasizes that conventional efforts that aim to recover muscle quantity will likely remain ineffective for regaining strength and improving quality of life after sepsis until deficiencies in muscle quality are addressed.
DOI: https://doi.org/10.7554/eLife.49920.001

**eLife digest** Sepsis is a life-threatening condition that occurs when a local infection spreads to the bloodstream and the body responds in such an exaggerated way that organs become damaged. Patients often require longs stays in intensive care units, and upon discharge experience chronic physical weakness and fatigue for several years. However, it was difficult to understand how sepsis can create these long-term problems because there was no way to study these issues in animals.

To fill this knowledge gap, Owen et al. developed a protocol where they triggered sepsis in adult mice and then used therapeutic treatments similar to the ones found in intensive care units; as a result, most of the animals survived, with many then exhibiting chronic muscle weakness. Further observations in surviving mice revealed that muscle mass recovered after sepsis, so this weakness was not due to a drop in muscle mass: instead, the quality of the muscle fibers had worsened. More specifically, there were striking abnormalities in mitochondria, structures whose role is to power cells. The muscles also showed signs of persistent oxidative damage, a process in which toxic molecules produced by life processes accumulate and end up harming cells.

Overall, these data suggest that reduced muscle quality contributes to chronic weakness after sepsis. While current programs for sepsis survivors aim to increase muscle quantity, the results by Owen et al. suggest that improving muscle quality, for example using antioxidant therapies, could be a new avenue of treatment.

DOI: https://doi.org/10.7554/eLife.49920.002

## Introduction

Sepsis is a common life-threatening condition caused by a deregulated host response to infection. This syndrome is characterized by profound systemic inflammation and disseminated intravascular coagulation, which often lead to multiple organ failure (MOF) and subsequent mortality (*Singer et al., 2016*). The incidence of sepsis has risen by 9% to 13% annually, largely due to an expansion of the elderly population, more frequent invasive surgical procedures, and increased antibiotic resistance (*Martin et al., 2003*; *Gaieski et al., 2013*; *Angus and Wax, 2001*; *Starr and Saito, 2014*). However, advances in critical care medicine and campaigns promoting early identification and treatment of sepsis have led to improved survival rates (*Angus and Wax, 2001*). Consequently, nearly 1.5 million sepsis survivors are discharged annually from U.S. hospitals, approximately 14 million globally (*Elixhauser et al., 2011*; *Prescott and Angus, 2018*; *Fleischmann et al., 2016*).

As the population of survivors grows, post-sepsis physical dysfunction, which exceeds that of intensive care unit acquired weakness (ICUAW) alone, has become a clear clinical problem (*Prescott and Angus, 2018*; *Callahan and Supinski, 2009*; *Iwashyna et al., 2010*; *Schefold et al., 2010*; *Contrin et al., 2013*). Sepsis survivors rarely return to baseline functional status after discharge from the ICU. Post-sepsis muscle weakness causes nearly half of previously functionally independent individuals to be discharged either to nursing care facilities or home with home care (*Odden et al., 2013*). Nearly a third of patients which were previously independent and had no prior comorbidities had problems with mobility one year after discharge (*Yende et al., 2016*). Further, survivors continue to develop functional limitations for at least 5 years following discharge, a trend that appears to be sepsis-specific (*Iwashyna et al., 2010*). Moreover, less than half of previously employed individuals are able to return to work within one year after discharge (*Poulsen et al., 2009*; *Pettilä et al., 2000*). Together these studies illustrate the dramatic impact of chronic post-sepsis weakness.

Although post-sepsis muscle weakness is now widely recognized as a serious medical issue, the lack of an appropriate animal model has greatly impeded the identification of mechanisms that contribute to *long-term* dysfunction. Current animal models of sepsis are either too severe, causing early death of most animals without recovery from sepsis, or too mild thus not triggering long-term chronic dysfunction. To overcome this issue, we recently refined a non-surgical murine model of polymicrobial sepsis whereby infection is initiated by injection of cecal slurry (CS) (*Starr et al., 2014*; *Starr et al., 2016*). Therapeutic intervention with a broad-spectrum antibiotic and fluids is provided, but initiated after bacteremia is evident (*Steele et al., 2017*). This delayed ICU-like resuscitation protocol allows for the development of sepsis with organ damage, yet rescues the majority of mice

from an otherwise completely lethal condition, thereby allowing the study of survivors. To further optimize our animal model for the current study, careful attention was also given to age, as the large majority of sepsis patients are late middle-age and older, and aging is an established risk factor for sepsis incidence, severity, and mortality (*Angus and Wax, 2001*; *Starr and Saito, 2014*; *Elixhauser et al., 2011*; *Martin et al., 2006*; *Dombrovskiy et al., 2007*).

The purpose of the present study was to establish that chronic muscle weakness, similar to the clinical condition among sepsis survivors, can be modeled in age-appropriate mice using our CS protocol with delayed ICU-like intervention. We then aimed to delineate underlying mechanisms responsible for post-sepsis muscle dysfunction. We show that sepsis survivors have significant skeletal muscle weakness for at least one month which cannot be attributed to muscle atrophy, but rather is associated with impaired mitochondrial activity and persistent protein oxidative damage.

## Results

### Mice exhibit chronic muscle weakness after sepsis induced by a severe model

We adapted our recently reported ICU-like model of sepsis to late middle-aged C57BL/6 mice (*Steele et al., 2017*) (16 months; equivalent to ~50-year-old human [*Flurkey K et al., 2007*]). Sepsis was induced by bolus injection of cecal slurry (CS) and therapeutic resuscitation with antibiotics and fluids was initiated at 12h and continued twice daily for five days (schematic provided in *Figure 1A*). This protocol rescued 74.1% of middle-aged males from otherwise completely lethal (LD$_{100}$) sepsis (*Figure 1B*, p<0.0001). No further mortality was observed after day 14. Assessment of bacteremia showed that resuscitation decreased bacterial load by day 2 (p=0.009), and resolved the systemic infection by day 4 (*Figure 1C*). Similar data were obtained using middle-aged female mice: 72.7% survival was achieved compared to 16% survival without therapeutic intervention (*Figure 1D*, p=0.035), and bacteremia was rapidly resolved (*Figure 1E*).

Next, we evaluated skeletal muscle strength in the murine sepsis survivors using ex vivo specific force analysis (muscle force normalized to physiological cross section) of the extensor digitorum longus hind limb muscle. Compared to non-sepsis controls (NSC), male sepsis survivors had a 24.6% reduction in maximal specific force at 2 weeks (p<0.0001) and 17.4% reduction at 1 month (p=0.002, *Figure 1F*). Similarly, female sepsis survivors were 19.8% weaker compared to controls at 2 weeks (p=0.035, *Figure 1G*). Sepsis-induced chronic muscle weakness could not be explained by atrophy since muscle force measurements are normalized to muscle size (physiological cross section). To exclude the possibility that reduced muscle function in sepsis survivors was due to repeated antibiotic administration, force was also measured in non-sepsis animals which received resuscitation with antibiotics and fluids in parallel to sepsis mice. We found that specific force was unchanged by resuscitation procedures (maximal force: 19.0 ± 0.8 N/cm2 in NSC vs 19.5 ± 0.8 N/cm$^2$ in NSC + resuscitation, p=0.359).

To determine if chronic muscle weakness was due in part to ongoing inflammation in the sepsis survivors, cytokine concentration in the plasma and gene expression in muscle were measured. Despite resolution of bacteremia, plasma IL-6 concentration was elevated at day 4 (p<0.0001), but comparable to NSC by 2 weeks in both male and female sepsis survivors (p=0.999 and p=0.667, respectively, *Figure 2A*). Plasma TNFα (*Figure 2B*) and IL-10 (*Figure 2C*) followed a similar trend. Likewise, gene expression of these cytokines in gastrocnemius muscle showed that pro-inflammatory IL-6 (*Figure 2D*) and TNFα (*Figure 2E*) were comparable among NSC and sepsis survivors at 2 weeks (p=0.622 and p=0.565, respectively). These data indicate that muscle weakness is not associated with ongoing systemic or local muscle inflammation. Interestingly, IL-10, an anti-inflammatory cytokine, was 2.5-fold higher in sepsis survivors compared to NSC (p=0.011, *Figure 2F*).

In addition, we observed splenomegaly in many sepsis animals on day 4 (males, 76.4% larger, p=0.0003 compared to NSC) which remained elevated at 2 weeks (males, 49.7% larger, p=0.004; females 20.2% larger, p=0.034), but recovered by 1 month (males p=0.905, *Figure 2G,H*) which may be indicative of high antigen clearance during recovery from sepsis (*Bronte and Pittet, 2013*). Taken together, these results demonstrate that middle-aged sepsis survivors exhibit chronic muscle weakness, even after bacteremia and systemic inflammation are resolved.

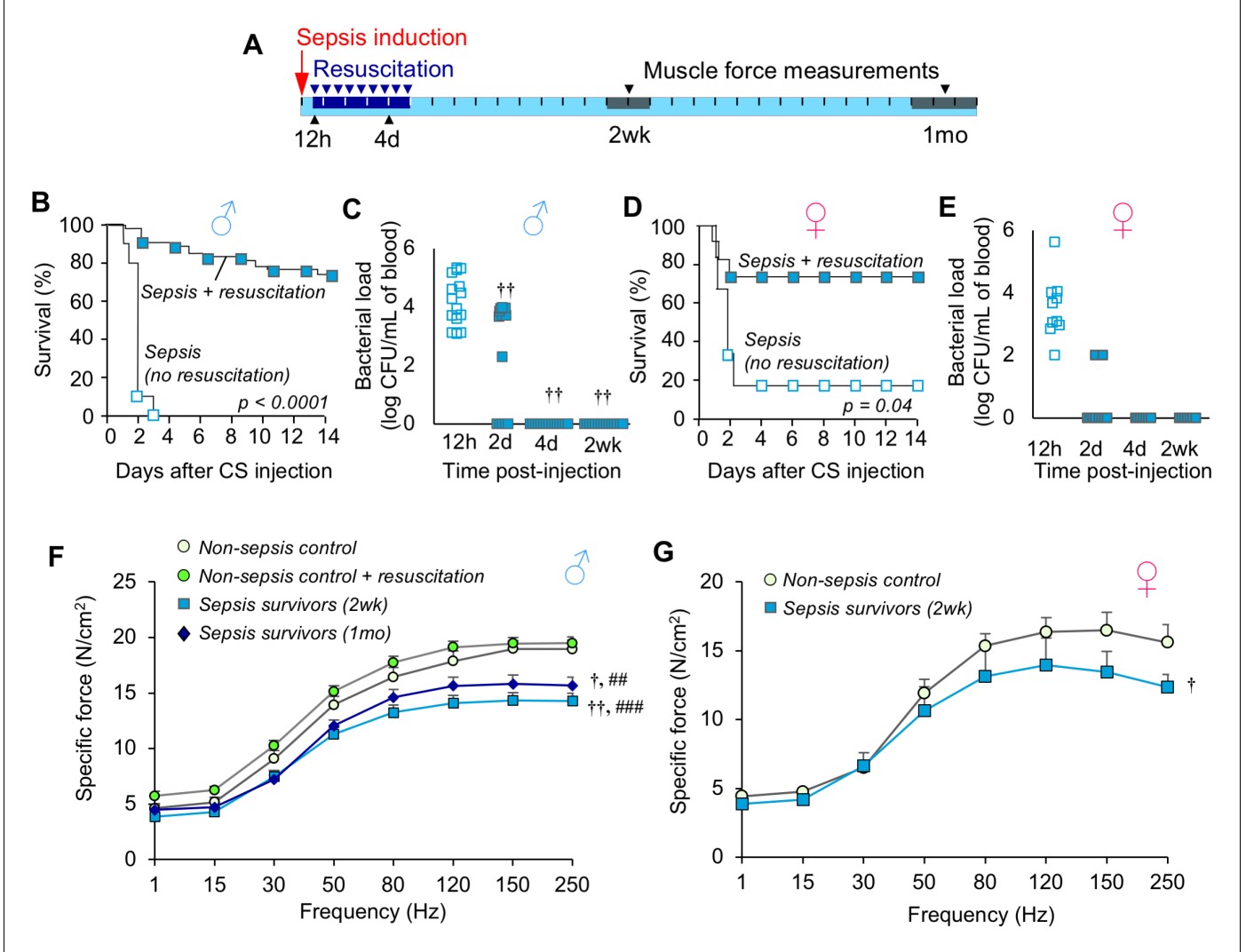

**Figure 1.** Mice exhibit chronic muscle weakness after recovering from sepsis induced by a severe model with ICU-like resuscitation. (A) Schematic diagram of the protocol which allows for long-term assessments in sepsis survivors. Sepsis is induced by cecal slurry (CS) injection (i.p.) and therapeutic resuscitation is delayed until 12h. Resuscitation includes antibiotics and fluid administration which is continued twice daily for 5 days. Each segment on the line corresponds to one day. (B, D) After sepsis induction, animals received either no intervention (n = 10 ♂, n = 6 ♀) or therapeutic resuscitation (n = 54 ♂, n = 11 ♀); survival was monitored for 14 days. Kaplan-Meier Log-rank test was performed. (C, E) Circulating bacterial load was assessed immediately before initiation of resuscitation (12h), and at 2d, 4d, and 2wk (n = 15 ♂, n = 7 ♀). Repeated measures ANOVA was performed (†† p<0.01 compared to 12h). (F, G) Specific force of the extensor digitorum longus was measured to assess muscle strength of non-sepsis control (NSC; n = 9 ♂, n = 5 ♀), NSC + resuscitation (n = 7 ♂), and sepsis surviving mice at 2wk (n = 7 ♂, n = 5 ♀) and 1mo (n = 7 ♂) after sepsis. Data are expressed as means ± SEM, † p<0.05, †† p<0.01 compared to NSC, ## p<0.01, ### p<0.001 compared to NSC + resuscitation.
DOI: https://doi.org/10.7554/eLife.49920.003

The following source data is available for figure 1:

**Source data 1.** Mice exhibit chronic muscle weakness after recovering from sepsis induced by a severe model with ICU-like resuscitation.
DOI: https://doi.org/10.7554/eLife.49920.004

## Experimental sepsis induces acute muscle wasting which recovers in murine sepsis survivors

Animals with sepsis lost significant body weight over time (males p=0.003 compared to NSC, *Figure 3A*; females p=0.001, *Figure 3—figure supplement 1*). Analyses of body composition by EchoMRI revealed a significant loss of fat mass after sepsis induction (p=0.004 compared to NSC, *Figure 3B*). On the other hand, loss of lean mass was observed during the first 5 days (p=0.005

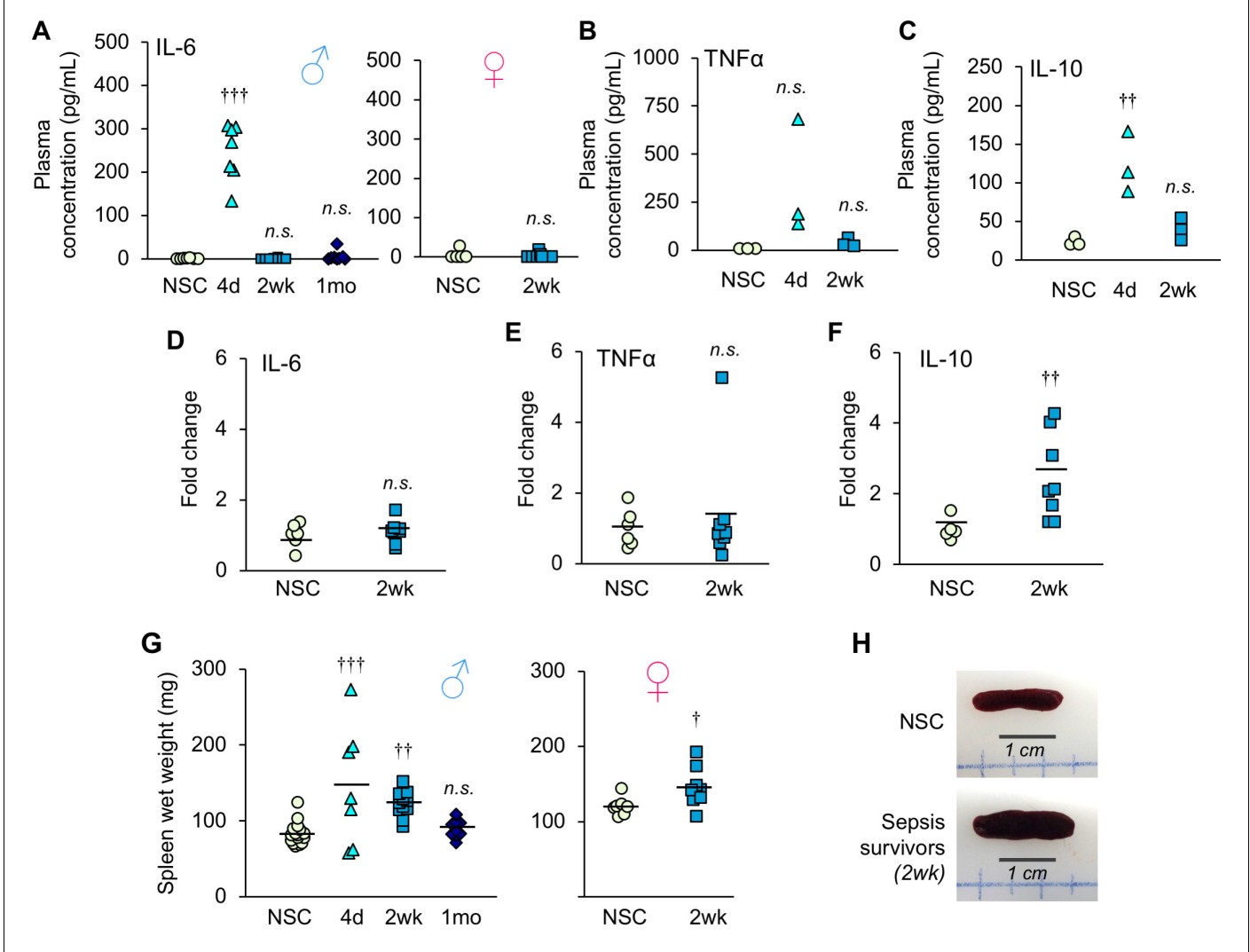

**Figure 2.** Inflammation is resolved by two weeks after sepsis. (A) Plasma IL-6 concentration was measured in non-sepsis control (NSC) and sepsis survivors (*n* = 7–8 ♂, *n* = 6–9 ♀, per group). (B) Plasma TNFα and (C) IL-10 concentrations were quantified in non-sepsis control (NSC), and sepsis surviving mice (*n* = 3 per group). Relative gene expression of (D) IL-6, (E) TNFα, and (F) IL-10 in gastrocnemius of male NSC (*n* = 6) and sepsis survivors (*n* = 8). (G) Spleen wet weight was recorded at time of sacrifice at 4d (*n* = 7 ♂), 2wk (*n* = 12 ♂, *n* = 9 ♀), and 1mo (*n* = 10 ♂) alongside non-sepsis controls (*n* = 16 ♂, *n* = 7 ♀). One-way ANOVAs were performed. *n.s.* not significant, † p<0.05, †† p≤0.01, ††† p<0.001 compared to NSC. (H) Representative macroscopic images of the spleen.

DOI: https://doi.org/10.7554/eLife.49920.005

The following source data is available for figure 2:

**Source data 1.** Inflammation is resolved by two weeks after sepsis.

DOI: https://doi.org/10.7554/eLife.49920.006

compared to NSC) which steadily recovered thereafter and became comparable to non-sepsis controls by two weeks (p=0.821, *Figure 3C*). In a selected experiment in which sepsis survivors were kept 1 month post-sepsis, body weight remained significantly lower than baseline (6.12% lower, p<0.0001, data not shown). These results suggest that sustained weight loss in murine sepsis survivors is attributable to loss of fat mass, not lean mass.

EchoMRI analysis allowed repeated measures of whole body composition without need to euthanize the animals; however, analysis of lean mass is not specific to skeletal muscle. Thus, we compared wet weight of hind limb skeletal muscles from male mice at day 4 (the time at which EchoMRI data showed the most profound loss of lean mass), 2 weeks, and 1 month, alongside non-sepsis

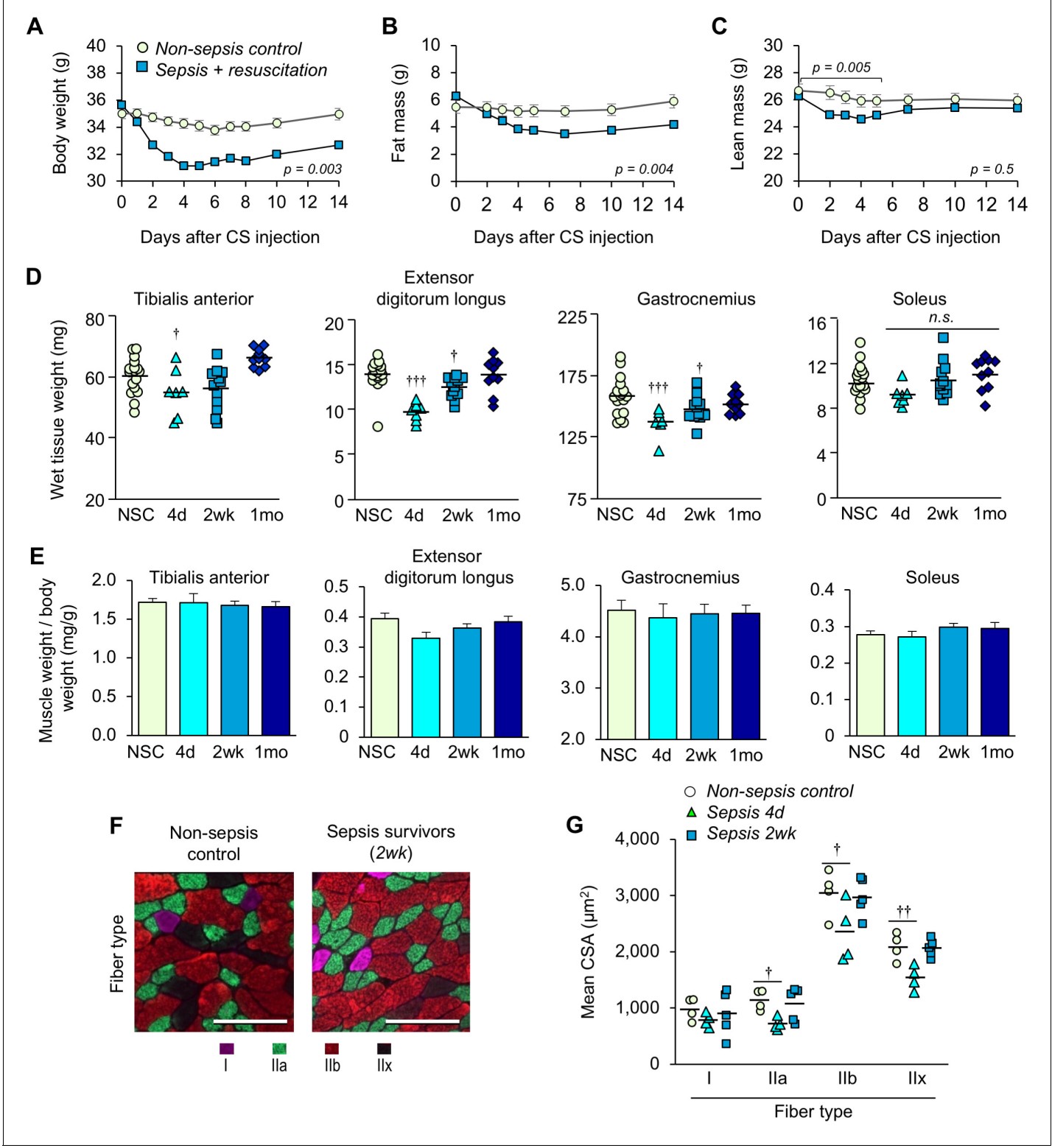

**Figure 3.** Sepsis-induced muscle wasting is evident during the acute phase but is later recovered. (A) Body weight, (B) fat mass, and (C) lean mass were assessed in non-sepsis controls (NSC, $n = 7$) alongside animals with sepsis + resuscitation ($n = 11$). Repeated Measures ANOVAs were conducted (A–C). (D) Wet tissue weight of hindlimb skeletal muscles was measured from NSC ($n = 17$) and sepsis mice at 4d, ($n = 7$), 2wk ($n = 12$), and 1mo ($n = 10$). One-way ANOVAs were performed. *n.s.* not significant, † $p<0.05$, ††† $p<0.001$ compared to NSC. (E) Skeletal muscle weight/body weight ratio. Data are presented as means ± SEM. One-way ANOVAs were performed, no statistical difference was found across the groups. Similar trends were observed

*Figure 3 continued on next page*

*Figure 3 continued*

in female sepsis survivors (*Figure 3—figure supplement 1*). (F) Representative images of fiber-type staining of the gastrocnemius show myosin heavy chain type I fibers (pink), IIa (green), IIb (red), and IIx (unstained/black); scale bars represent 100 µm. (G) Cross-sectional area (CSA) analysis by fiber type (*n* = 4–5 per group). Staining and CSA analysis on soleus muscles are provided in *Figure 3—figure supplement 2*. One-way ANOVAs were performed, † p<0.05, †† p<0.01, compared to NSC.

DOI: https://doi.org/10.7554/eLife.49920.007

The following source data and figure supplements are available for figure 3:

**Source data 1.** Sepsis-induced muscle wasting is evident during the acute phase but is later recovered.

DOI: https://doi.org/10.7554/eLife.49920.010

**Figure supplement 1.** Sepsis-induced body weight and skeletal muscle weight in female animals.

DOI: https://doi.org/10.7554/eLife.49920.008

**Figure supplement 2.** Sepsis-induced atrophy is not evident in soleus muscle.

DOI: https://doi.org/10.7554/eLife.49920.009

controls (*Figure 3D*). The wet weight of the predominantly fast-glycolytic muscles tibialis anterior (TA) and extensor digitorum longus (EDL), and the mixed fiber-type gastrocnemius (GA) were reduced at day 4 compared to non-sepsis controls (p=0.054, p<0.0001, and p=0.001, respectively). The wet weight of these muscles was lower at 2 weeks and was comparable to non-sepsis controls at 1 month (p=0.918 for EDL and p=0.242 for GA). The weight of the predominantly oxidative soleus muscle was similarly reduced at day 4, but did not achieve statistical significance (p=0.055 across groups). While these data are indicative of muscle loss, it is important to note that changes in muscle size were proportional to loss of body weight as evidenced by skeletal muscle weight to body weight ratio (*Figure 3E*). Such trends in body weight, muscle wet weight, and muscle weight to body weight ratio were similarly observed in female sepsis survivors (*Figure 3—figure supplement 1*).

To more robustly assess atrophy, and to evaluate potential fiber-type differences in the post-sepsis condition, we performed myofiber-specific cross-sectional area (CSA) analysis on the medial head of the mixed fiber-type gastrocnemius (*Figure 3F,G*) muscle. The mean CSA of fast-twitch (type II) fibers showed sepsis-mediated atrophy on day 4 (IIa p=0.023, IIb p=0.044, and IIx p=0.0034) which was largely recovered by 2 weeks, whereas little, if any, atrophy was observed in slow (type I) fibers (p=0.626). Analysis of the predominantly slow-oxidative soleus muscle did not reveal sepsis-induced atrophy (type I fibers p=0.521, IIa p=0.799, *Figure 3—figure supplement 2*) which is consistent with the wet weight (*Figure 3D*, right panel). These data are consistent with clinical reports which show that preferential atrophy of fast-twitch fibers occurs during critical illness (*Gutmann et al., 1996*; *Bierbrauer et al., 2012*).

Collectively, these data provide evidence that chronic muscle weakness in murine sepsis survivors cannot be attributed to muscle wasting alone. Therefore, a long-term reduction in muscle function of murine sepsis survivors must be largely attributable to impaired quality, rather than quantity.

## Profound morphological defects are evident in skeletal muscle mitochondria from murine sepsis survivors

In sepsis patients, skeletal muscle mitochondrial dysfunction is evident during their stay in the ICU (*Fredriksson et al., 2006*; *Fredriksson et al., 2008*), but it is unknown whether such mitochondrial derangements persist after recovery from sepsis. Therefore, we assessed mitochondrial integrity in the EDL of sepsis survivors 2 weeks after CS injection. Transmission electron microscopy was used to study intermyofibrillar (IMF) mitochondria (*Figure 4A*) and subsarcolemmal (SS) mitochondria (*Figure 4B*). Normal mitochondrial morphology was evident in non-sepsis controls characterized by highly organized lamellar cristae (i.e.inner mitochondrial membrane arranged in parallel stacks). On the other hand, striking morphological alterations were observed frequently in both IMF mitochondria and SS mitochondria in muscle from sepsis survivors. Even observation at low magnification (5,000X) revealed numerous mitochondria with fragmented cristae and enlarged matrix space in the muscles from sepsis survivors. These features are better appreciated at higher magnification (15,000X) by which we also observed some mitochondria nearly devoid of cristae, 'onion-like' concentric swirling of cristae (*Jiang et al., 2017*; *Walker and Benzer, 2004*), as well as compartmentalization in vesicle-like structures (*Sun et al., 2007*; *Vincent et al., 2016*). Further, observation of

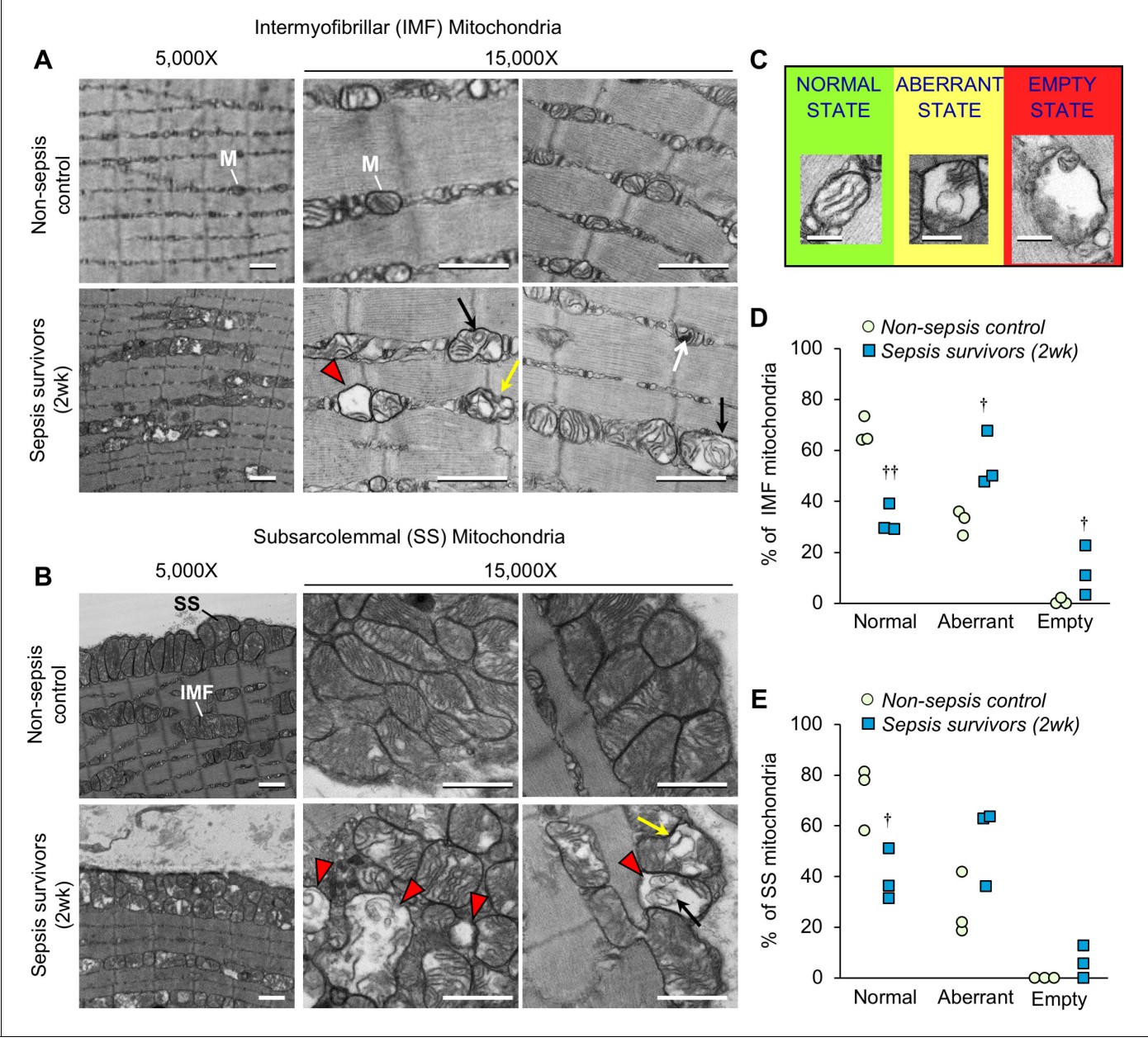

**Figure 4.** Sepsis triggers profound long-lasting ultrastructural defects in skeletal muscle mitochondrial populations. The extensor digitorum longus was harvested from murine sepsis survivors (2wk) and non-sepsis controls (*n* = 3 per group) and processed for transmission electron microscopy observation of mitochondria (M). Representative micrographs of (A) intermyofibrillar (IMF) and (B) subsarcolemmal (SS) mitochondria at 5,000X and 15,000X magnifications are shown (scale bars represent 1,000 nm). Abnormal mitochondrial structures are indicated with the following symbols: destruction of cristae with expanded matrix space (*red arrowheads*), concentric 'onion shaped' cristae (*black arrows*), compartmentalization into vacuolar structures (*yellow arrows*), and densely compacted cristae (*white arrows*). (C) Representative images for classification of normal, aberrant, or empty mitochondria for morphometric analyses of (D) intermyofibrillar and (E) subsarcolemmal mitochondrial populations which are shown as percent for non-sepsis control and sepsis survivors (2wk). Repeated-measures ANOVAs were performed, each considering a two-way interaction between group and category (normal/aberrant/empty). *n.s.* not significant, † $p<0.05$, †† $p<0.01$ compared to NSC.

DOI: https://doi.org/10.7554/eLife.49920.011

The following source data and figure supplement are available for figure 4:

**Source data 1.** Sepsis triggers profound long-lasting ultrastructural defects in skeletal muscle mitochondrial populations.

DOI: https://doi.org/10.7554/eLife.49920.013

**Figure supplement 1.** Mitochondrial ultrastructure in tibialis anterior skeletal muscle of murine sepsis survivors.

DOI: https://doi.org/10.7554/eLife.49920.012

mitochondrial ultrastructure in TA muscles of the sepsis survivors (*Figure 4—figure supplement 1*) showed abnormalities consistent with those found in the EDL.

In an effort to better understand these marked ultrastructural abnormalities in the muscles of sepsis surviving mice, we performed morphometric analysis. Five representative images were acquired for each mitochondrial population (IMF and SS). On average, 49 IMF and 69 SS mitochondria were observed and categorized in muscles from three animals per group. Normal mitochondria were defined as having intact cristae which occupied ≥80% of the mitochondrial area and had no observable compartmentalization; aberrant cristae were defined as cristae which occupied 20–80% of the mitochondrial space and/or had indications of swirled cristae or compartmentalization; 'empty' mitochondria were defined as exceptionally damaged whereby < 20% of the mitochondrial space was devoid of cristae and/or the outer mitochondrial membrane was seemingly ruptured and accompanied by enlargement (representative examples are provided in *Figure 4C*). Normal mitochondria made up a significantly smaller proportion of the total IMF (*Figure 4D*) and SS (*Figure 4E*) mitochondria in sepsis survivors (p=0.002 and p=0.048, respectively). IMF mitochondria were seemingly more affected in the post-sepsis condition, where aberrant and empty mitochondria represented a significantly larger proportion of the mitochondria (p=0.029 and 0.049, respectively) compared to controls. This trend was similar in the SS mitochondrial population, but did not achieve statistical significance.

## Sepsis triggers long-term reductions in skeletal muscle mitochondrial bioenergetics

Since we observed altered mitochondrial morphology post-sepsis, we hypothesized that energy metabolism would be impaired. Thus, mitochondria were isolated from the TA of mice 2 weeks after sepsis and oxygen consumption rates (OCR) were measured to assess mitochondrial bioenergetics in the post-sepsis condition. These results showed that the maximal ADP phosphorylation rate (State III) and Complex I-driven electron transport (State V-CI) were significantly lower (27.3% *p*=0.003, and 26.6% *p*=0.017, respectively) in sepsis survivors compared to that of controls (*Figure 5*). On the other hand, Complex II-driven maximum electron transport (State V-CII) showed a non-significant (p=0.099) trend of lower OCR in sepsis survivors (*Figure 5*). These results show that skeletal muscle mitochondria in sepsis survivors have significantly reduced mitochondrial bioenergetics.

Since acute sepsis affects food consumption, we confirmed that the observed long-term changes in mitochondrial bioenergetics was not influenced by reduced food intake. After measuring baseline food consumption for 5 days, cecal slurry was injected and food consumption was monitored daily for 2 weeks (*Figure 5—figure supplement 1A*). On the first day of sepsis, animals ate only 11.08% of their baseline (0.52 g compared to 5.19 g; p<0.001), which steadily increased to 75.51% by day 4 (3.53 g, p=0.042) and returned to near-baseline thereafter. To mimic this reduced food consumption, a pair-feeding paradigm was followed by which the daily food intake of a set of non-sepsis animals

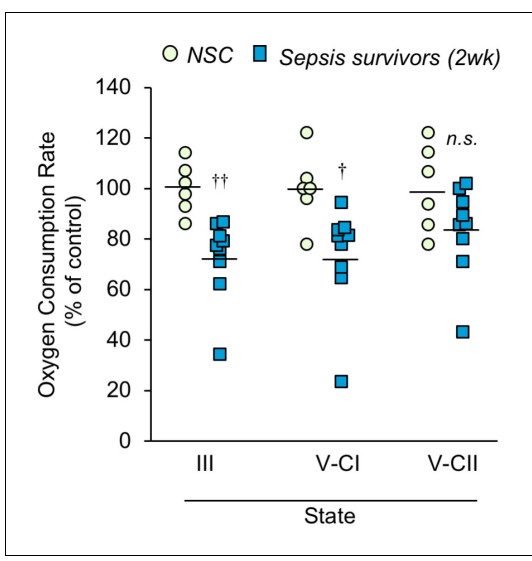

**Figure 5.** Mitochondrial respiration capacity is reduced in skeletal muscle of sepsis survivors. Mitochondria were isolated from the tibialis anterior of non-sepsis control (NSC, *n* = 6) and sepsis survivors (2wk, *n* = 9) and oxygen consumption rate was measured in triplicate for each sample. State III refers to the ADP phosphorylation rate. States V-CI and V-CII refer to complex I and complex II driven maximum electron transport, respectively. Data are represented as individual data points with mean values shown. Two-sample t-tests were performed. *n.s.* not significant, † p<0.05, †† p<0.01 compared to NSC.
DOI: https://doi.org/10.7554/eLife.49920.014

The following source data and figure supplement are available for figure 5:

**Source data 1.** Mitochondrial respiration capacity is reduced in skeletal muscle of sepsis survivors.
DOI: https://doi.org/10.7554/eLife.49920.016

**Figure supplement 1.** Mitochondrial respiration is unchanged in pair-fed food-restricted mice.
DOI: https://doi.org/10.7554/eLife.49920.015

was restricted to match the daily food consumption by the sepsis animals. Ad libitum (freely-fed) non-sepsis mice were included as controls. Unsurprisingly, pair feeding resulted in significant reductions in body weight over time (p<0.001; *Figure 5—figure supplement 1B*). The wet weight of hindlimb skeletal muscles was lower in the pair-fed mice than freely-fed controls (GA p<0.001, TA p=0.004; *Figure 5—figure supplement 1C*), however the reduction was proportional to overall loss of body weight (GA p=0.151, TA p=0.813; *Figure 5—figure supplement 1D*), similar to the trend observed in the sepsis survivors (*Figure 3*). Oxygen consumption rate (OCR) was measured in isolated mitochondria from the TA which showed similar mitochondrial respiration capacity among pair-fed and freely-fed mice (State III p=0.944, State V-CI p=0.737, State V-CII p=0.185; *Figure 5—figure supplement 1E*). These data confirm that the reduced food intake during sepsis does not contribute to chronic mitochondrial impairment in sepsis surviving mice.

Although OCR of isolated mitochondria is widely used to assess functional capacity, the severity of the phenotype is potentially masked because severely damaged mitochondria are lost during isolation. In addition, OCR reflects the maximum capacity in the presence of excess amounts of substrates. In contrast, the efficiency of substrate utilization, which does not influence OCR, is also a major determinant of mitochondrial activity. Therefore, we also performed a series of histochemical stains on tibialis anterior muscle sections as an alternative method to evaluate specific mitochondrial complex enzyme activity in whole tissue (*Figure 6A*). Quantification of stain intensities was conducted (*Figure 6B*) which showed that complex I activity (measured by nicotinamide adenine dinucleotide dehydrogenase (NADH), *Figure 6A*, *top row*) was 22.2% lower at day 4% and 40.8% lower at 2 weeks compared to NSC, although statistical significance was not achieved. Complex II activity (measured by succinate dehydrogenase (SDH), *Figure 6A*, *middle row*) was 19.8% lower at day 4% and 38.1% lower at 2 weeks (p=0.015). Overall mitochondrial respiration activity was assessed by cytochrome c oxidase staining (COX; complex IV; *Figure 6A*, *bottom row*) which was 43.7% lower at day 4 (p<0.001) and 48.6% lower at 2 weeks (p<0.001). Interestingly, 2 weeks after sepsis some myofibers had areas devoid of staining which was consistent among serial sections for the different mitochondrial enzyme activities (*Figure 6*, *right column, yellow arrows*). Using hematoxylin and eosin stained serial sections we confirmed that the areas devoid of mitochondrial enzyme activity were not devoid of tissue. We hypothesize that the areas devoid of histochemical staining in the myofibers of sepsis survivors (*Figure 6A*, *right column*) are likely the areas that house 'empty' or highly 'aberrant' mitochondria as depicted in *Figure 4* by TEM analysis.

Taken together, these data demonstrate that mitochondrial complex enzyme activities are impaired long after sepsis itself is resolved. Complex I-driven respiration seems more affected than Complex II-driven activity as evidenced by respiration analysis in isolated mitochondria (*Figure 5*). However, histochemical analysis, which reflects mitochondrial bioenergetics in whole tissue, indicated that CII-driven activity is also significantly affected in the post-sepsis condition (*Figure 6*).

## Oxidative damage persists in skeletal muscle of sepsis survivors long after recovery from acute illness

Significant production of reactive oxygen species (ROS) and reactive nitrogen species (RNS) is triggered during sepsis due to inflammation, tissue hypoxia (*Ueda et al., 2008*; *Starr et al., 2011*), and mitochondrial dysfunction (*Fredriksson et al., 2006*; *Zolfaghari et al., 2015*; *Singer, 2014*; *Castello et al., 2006*). To investigate oxidative damage as a potential contributor to muscle weakness in the post-sepsis condition, whole protein was extracted from the tibialis anterior of control and post-sepsis animals (both at 2 weeks and 1 month), and markers of irreversible oxidative damage were assessed by Western blot. Carbonylation was detected in proteins ranging in size from ~37 to 250 kDa, and showed ~2 fold higher levels in sepsis survivors at 2 weeks, as quantified by densitometric analysis of the entire lane (p=0.002, *Figure 7A*). This trend was no longer evident at 1 month (p=0.832, *Figure 7B*). Multiple bands were also detected with the oxidative marker 3-nitrotyrosine, which was ~2.5 fold higher in muscle from 2 week sepsis survivors (p=0.016, *Figure 7C*) which remained modestly elevated at 1 month (p=0.099, *Figure 7D*). These results demonstrate that sepsis survivors have significant oxidative and nitro-oxidative damage even long after the acute sepsis condition has passed, which may indicate that (1) damage is not successfully cleared and therefore persists in the post-sepsis condition, and/or (2) damage is potentially propagated continuously by impaired mitochondrial activity.

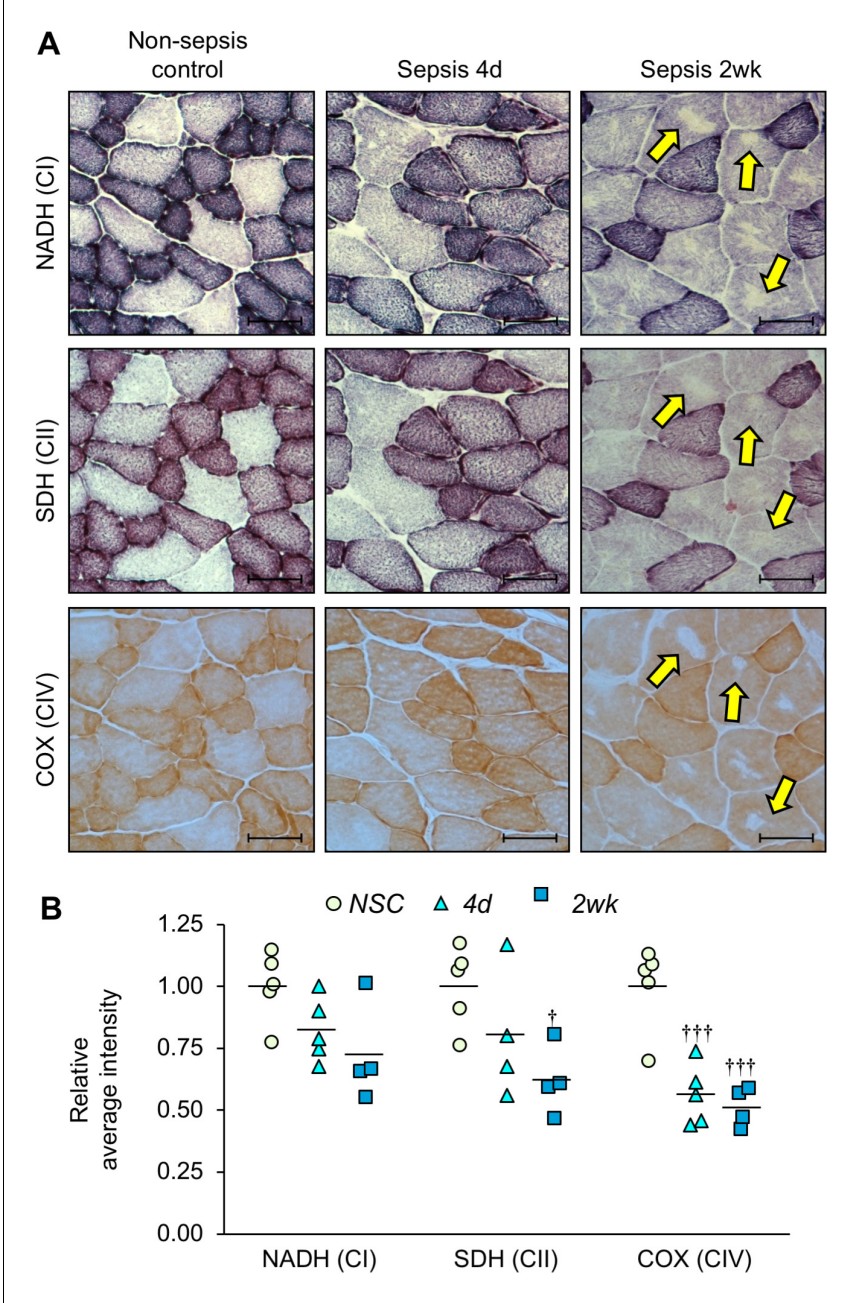

**Figure 6.** Mitochondrial enzyme activities are progressively reduced over time in sepsis survivors. Tibialis anterior specimens were utilized for histochemical staining of oxidative phosphorylation enzyme activities in non-sepsis controls (*n* = 5), and after sepsis at 4d (*n* = 4–5) and 2wk (*n* = 4). (**A**) Representative images of staining for nicotinamide adenine dinucleotide dehydrogenase (NADH; complex I, *top*), succinate dehydrogenase (SDH; complex II, *middle*), and cytochrome C oxidase (COX; complex IV, *bottom*) enzyme activities conducted on serial sections. Scale bars represent 50 μm; arrows indicate areas devoid of mitochondrial enzyme activity and correspond to the same fiber in serial sections. (**B**) Average intensities were quantified using Aperio ImageScope software and normalized to the intensity of the controls. Data were analyzed by one-way ANOVA. † p<0.05, ††† p<0.001 compared to NSC.

DOI: https://doi.org/10.7554/eLife.49920.017

The following source data is available for figure 6:

**Source data 1.** Mitochondrial enzyme activities are progressively reduced over time in sepsis survivors.
DOI: https://doi.org/10.7554/eLife.49920.018

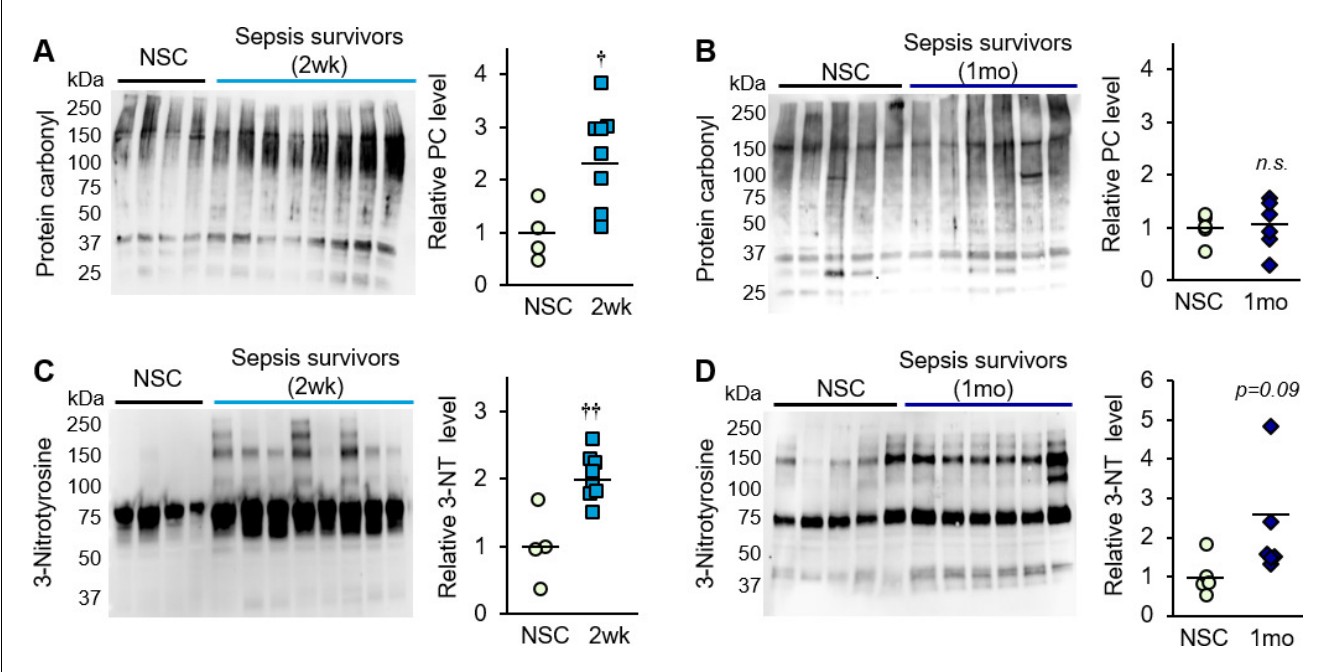

**Figure 7.** Oxidative damage is evident in skeletal muscle of sepsis survivors. Protein isolated from whole tibialis anterior muscles of non-sepsis control (NSC) and sepsis survivors was used to assess markers of oxidative damage by Western blot. Protein carbonylation (PC) and nitrotyrosination (3-NT) were detected in NSC ($n = 4$–5), and sepsis survivors at (**A, C**) 2 weeks ($n = 8$) and (**B, D**) 1 month ($n = 6$). Uncropped blots are shown. Total intensity of each lane was quantified by densitometric analysis and normalized to total protein content. Data are expressed as individual data points with mean values shown. Two-sample t-tests were performed. *n.s.* not significant, † $p < 0.05$, †† $p < 0.01$ compared to NSC.

DOI: https://doi.org/10.7554/eLife.49920.019

The following source data is available for figure 7:

**Source data 1.** Oxidative damage is evident in skeletal muscle of sepsis survivors.

DOI: https://doi.org/10.7554/eLife.49920.020

## Discussion

Clinically, severe persistent physical impairment in sepsis survivors is a well-recognized phenomenon; however, progress towards understanding the underlying mechanisms have been limited largely due to the lack of an appropriate animal model with which to study long-term effects. Here, we describe our refinements to a murine model of sepsis, which enabled us to induce severe pathogenesis (i.e. with multi-organ damage and high risk of mortality), yet rescue the majority of mice to obtain survivors in which long-term muscle function could be assessed. To the best of our knowledge, this is the first study to demonstrate skeletal muscle dysfunction long after recovery from experimental sepsis. We show that murine sepsis survivors have prolonged muscle weakness, independent of muscle atrophy, that is associated with ongoing mitochondrial dysfunction and oxidative damage.

In the refinement of our model, we addressed four specific areas that were overlooked in prior research: (*Singer et al., 2016*) age of animals, (*Martin et al., 2003*) sepsis severity, (*Gaieski et al., 2013*) inclusion of muscle strength analysis, and (*Angus and Wax, 2001*) avoiding artificial effects of sepsis model. *First*, in addition to being more likely to develop sepsis (*Starr and Saito, 2014*; *Elixhauser et al., 2011*; *Dombrovskiy et al., 2007*), patients at late middle-age more frequently suffer from post-sepsis dysfunction due to confounding issues such as preexisting comorbidities, reduced muscle mass, and lower protein intake (*Contrin et al., 2013*; *Rahman et al., 2014*). Further, these patients are most vulnerable to economic repercussions since post-sepsis functional limitations often prevent them from returning to work (*Iwashyna et al., 2012*; *Hofhuis et al., 2008*). *Second*, in order to observe long-term muscle weakness, considerably *severe* sepsis needed to be induced. Battle and colleagues (*Battle et al., 2014*) reported that patients who survived septic shock had reduced physical functioning and general health compared to patients who survived uncomplicated

sepsis or sterile (i.e. non-infectious) systemic inflammatory response syndrome. Likewise, this suggests that animal models of sepsis with mild or modest severity (i.e. no to little mortality in the absence of resuscitation) have little clinical relevance. In order to model the morbidities with which sepsis survivors present clinically, experimental sepsis should be severe enough to cause chronic dysfunction, as we demonstrate in the present study. *Third*, the limited research that has been conducted at later time-points after experimental sepsis did not include muscle function assessment in parallel with molecular analyses, thus actual muscle weakness was not confirmed. One exception, *Rocheteau et al. (2015)* assessed muscle force (specific tension) in mice 21 days after injury induced by a combination of myotoxin injection and sepsis (cecal ligation and puncture; CLP); however, it is unclear whether sepsis itself had any effect on muscle force and no comparison was made with non-sepsis controls. *Fourth*, many sepsis studies use the CLP animal model; however, in this model surgical complications may influence recovery. Further, differences in cecum shape and size among older mice lead to inconsistency in ligation site (*Starr et al., 2014*), and in mice kept long-term, the ligated cecum may result in unresolved necrosis, and sustained inflammation.

In an effort to better model the clinical course of sepsis by addressing the above four factors, we induced sepsis by CS injection in late middle-aged mice, and used our ICU-like severe model of sepsis (*Steele et al., 2017*) which resulted in ~75% survival after an otherwise lethal insult. In the current study, using ex vivo force analysis, we found that middle-aged sepsis surviving mice exhibit a ~ 20% reduction of muscle force compared to non-sepsis controls up to 1 month after sepsis. This ex vivo muscle strength assessment circumvents potential confounding variables, such as behavioral, sensory, and operational factors, which are introduced by the commonly-used grip strength measurement (*Maurissen et al., 2003*). We then provide clear evidence that long-term muscle weakness cannot be explained by loss of muscle mass, but rather reduced muscle quality. We show that specific force, which is normalized to unit area (i.e. muscle size), is significantly reduced in murine sepsis survivors. These data imply that differences in muscle size among control and sepsis surviving animals did not account for differences in muscle strength. We additionally provide body composition analysis, wet weights of skeletal muscles, and myofiber cross-sectional area which collectively demonstrate that animals indeed have atrophy during sepsis, but that muscle mass largely recovers. Thus, this model mimics the clinical situation since aged subjects are most at risk for sepsis incidence, morbidity, and mortality. Further, it models non-surgical sepsis, and development of severe infection and inflammation prior to therapeutic intervention, similar to the typical clinical scenario. Additionally, this long-term model replicates the clinical presentation of post sepsis syndrome, or the broader issue of post-intensive care syndrome (PICS), in which the patient survives but remains physically weak, even in the absence of muscle wasting.

This differentiation between quality vs quantity is highly important, as prevention of muscle wasting is a large clinical focus (*Cohen et al., 2015*). Muscle wasting is a major concern for sepsis patients in the ICU. Numerous factors, most notably muscle unloading during bedrest, insufficient nutrition, pharmacologic exposures, and imbalanced protein catabolism and synthesis, have been shown to contribute to sepsis-mediated atrophy (*Schefold et al., 2010*; *Schreiber et al., 2018*; *Eikermann et al., 2006*; *Spranger et al., 2003*; *Dres et al., 2017*; *Yu et al., 2018*; *Lang et al., 2007*; *Yang et al., 2018*; *Vary and Kimball, 1992*; *Puthucheary et al., 2013*; *Bouglé et al., 2016*). However, clinical interventions with early mobilization (*Denehy et al., 2013*; *Morris et al., 2016*; *Moss et al., 2016*; *Tipping et al., 2017*) and nutritional support (*Goossens et al., 2017*; *Hermans et al., 2013*; *Ogilvie and Larsson, 2014*) have not improved long-term outcomes, and some studies even show negative effects of high protein delivery on muscle wasting (*Puthucheary et al., 2013*). Beyond this key issue, muscle weakness is not always accompanied by muscle wasting: they are separate entities that should not be considered synonymously (*Schefold et al., 2010*; *Reid and Moylan, 2011*). However, in future studies it would be of interest to expand our experimental model to include other factors that are common for patients in the ICU, such as corticosteroid treatment, immobilization, and mechanical ventilation, which may further impact recovery and potentially reproduce the phenomenon of long-term muscle wasting post-sepsis. Nonetheless, taken together our data suggest therapeutic targets beyond muscle wasting are more likely to improve post-sepsis muscle weakness.

In this study, we demonstrate that mitochondrial damage may be a key driver of chronic muscle weakness in murine sepsis survivors. Healthy mitochondria are critical for efficient muscle function, an ATP-dependent process. Skeletal muscles are mitochondrial dense and heavy oxygen consumers

and thus greatly susceptible to damage during hypoxic conditions, including sepsis (*Zhou et al., 2014*). Using transmission electron microscopy, we demonstrate that mitochondria in sepsis survivors have striking morphological abnormalities, including absent or fragmented cristae, enlarged matrix spaces, vacuolar/compartmentalized structures, and concentric 'onion-like' swirling cristae. Notably, the IMF mitochondria, which are primarily responsible for ATP production for muscle contraction (*Hood, 2001*; *Timpani et al., 2015*), are more significantly damaged than SS mitochondria. As has been described (*Vincent et al., 2016*), mitochondrial structure and function are tightly linked; thus, derangements are associated with profound physiological implications. The loss of inner mitochondrial membrane (where oxidative phosphorylation occurs), as well as the abnormal presence of concentric/swirling cristae likely cause reduced bioenergetics in the post-sepsis condition.

Indeed we demonstrate that mitochondrial bioenergetics are impaired in murine sepsis survivors long after sepsis itself is resolved as shown using respiration analysis on isolated mitochondria and complex enzyme activity analyses on whole tissue. We also unexpectedly found that in some myofibers there are areas devoid of any functional mitochondrial enzyme activity, where mitochondria are not capable of converting energy substrates to detectable levels. Measuring glycolytic rates and TCA cycle flux may increase our understanding of mitochondrial bioenergetics in the post-sepsis condition which may be assessed using a metabolomics approach in future studies. As muscle contraction is dependent on ATP for cross-bridge cycling, one can conclude that such mitochondrial impairments would contribute to reduced muscle contraction. It is important to note that accumulation of such grossly damaged and dysfunctional mitochondria in middle-aged sepsis survivors is likely perpetuated by the fact that mitophagy decreases during aging (*Sun et al., 2007*; *Jang et al., 2018*). We are currently investigating mitophagy and mitochondrial turnover in murine sepsis survivors to observe if reduced mitophagy and/or biogenesis contribute to mitochondrial dysfunction.

Damaged mitochondria not only result in reduced ATP synthesis, but also in increased oxidative stress. As a byproduct of oxidative phosphorylation, mitochondria generate superoxide anions and nitric oxide. Mitochondrial damage, especially to complex I (*Castello et al., 2006*), triggers persistent production of reactive oxygen species (ROS) and reactive nitrogen species (RNS) (*Balaban et al., 2005*). Previous studies have shown that sarcomeric proteins, including tropomyosin, actin, and creatine kinase, scavenge free radicals during oxidative stress which causes post-translational modifications (*Fedorova et al., 2009*). We observed significant oxidative and nitro-oxidative damage in the muscle homogenates from sepsis survivors. Importantly, these oxidative modifications are irreversible (*Cai and Yan, 2013*) and require protein turnover for elimination. As oxidative stress has been shown to directly cause muscle fatigue (*Shindoh et al., 1990*; *Reid, 2008*; *Stasko et al., 2013*), our data suggest that persistent oxidative stress in the post-sepsis condition likely contributes to chronic muscle weakness. Moreover, the ROS and RNS secreted as a result of sepsis-induced mitochondrial damage then likely propagate mitochondrial dysfunction and energy failure through inhibition of the electron transport chain, thus acting as a vicious cycle of damage and functional decline (*Harman, 1972*; *Brandt et al., 2017*). Future proteomic analyses would allow us to further elucidate mechanisms of mitochondrial and sarcomeric protein damage, as well as uncover many other pathways of interest that may underlie chronic muscle weakness after critical illness. This is likely multiphasic due to different mechanisms during the acute and chronic state, and influenced by age, and thus requires careful investigation.

In summary, the present study demonstrates that profound weakness is present in the skeletal muscle of murine sepsis survivors and that such muscle weakness is not attributed to loss of muscle quantity alone, but rather is characterized by impaired quality on the mitochondrial and myofibrillar levels. Significant mitochondrial damage and dysfunction, as well as marked oxidative damage to skeletal muscle proteins, together likely contribute to chronic muscle weakness in sepsis survivors. We propose that mitochondrial dysfunction is central to other altered muscle physiology in the post-sepsis state, such as redox balance, calcium homeostasis, repressed autophagy, and mitochondrial permeability transition pore opening, altogether largely contributing to post-sepsis muscle weakness. This work provides strong evidence that therapeutic strategies aimed at restoring mitochondrial health, in addition to restoring muscle mass, will likely allow survivors to regain strength and improve quality of life after sepsis.

# Materials and methods

## Key resources table

| Reagent type (species) or resource | Designation | Source or reference | Identifiers | Additional information |
|---|---|---|---|---|
| Genetic Reagent (*M. musculus; male and female*) | C57BL/6 mice | National Institute on Aging | RRID:SCR_007317 | 16 months old |
| Biological sample (*M. musculus*) | Cecal slurry; CS | *Starr and Saito, 2014*; *Steele et al., 2017* | | Donor mice: 16 week old male C57BL/6 mice obtained from The Jackson Laboratory |
| Antibody | Mouse monoclonal myosin heavy chain type I (MIgG2b) | Developmental Studies Hybridoma Bank | Cat# BA-D5; RRID: AB_2235587 | 1:100 |
| Antibody | Mouse monoclonal myosin heavy chain type IIa (MIgG1) | Developmental Studies Hybridoma Bank | Cat# SC-71; RRID: AB_2147165 | 1:100 |
| Antibody | Mouse monoclonal myosin heavy chain type IIb (MIgM) | Developmental Studies Hybridoma Bank | Cat# BF-F3; RRID: AB_2266724 | 1:100 |
| Antibody | Goat anti-mouse IgG2b, Alexa Fluor 647 conjugated | Life Technologies-Thermo Fisher | Cat# A-21242; RRID: AB_2535811 | 1:250 |
| Antibody | Goat anti-mouse IgG1, Alexa Fluor 488 conjugated | Life Technologies-Thermo Fisher | Cat# A-21121; RRID: AB_2535764 | 1:500 |
| Antibody | Goat anti-mouse IgM, Alexa flour 555 conjugated | Life Technologies-Thermo Fisher | Cat# A-21426; RRID: AB_2535847 | 1:250 |
| Antibody | Mouse monoclonal 3-Nitrotyrosine (39B6) | Abcam | Cat# ab61392; RRID: AB_942087 | 1:3000 |
| Antibody | Goat anti-mouse IgG-HRP | Santa Cruz | Cat# SC-2005 | 1:10,000 |
| Commercial assay or kit | High sensitivity IL-6 mouse ELISA kit | eBIOSCIENCES-Thermo Fisher | Cat# BMS603HS; RRID: AB_2575654 | Sepsis 4d plasma samples diluted 5X for assay |
| Commercial assay or kit | V-PLEX multiplex assay | V-Plex Proinflammatory Panel 1 Mouse Kit, customized | Cat# K15048D | |
| Commercial assay or kit | PureLink RNA Mini Ki | Invitrogen-Thermo Fisher | Cat# 12183025 | |
| Commercial assay or kit | SuperScript III First-Strand Synthesis SuperMix | Life Technologies-Thermo Fisher | Cat# 18080400 | |
| Commercial assay or kit | TaqMan mouse IL-6 gene expression assay | Thermo Fisher | Assay ID: Mm00446190_m1 | |
| Commercial assay or kit | TaqMan mouse TNFα gene expression assay | Thermo Fisher | Assay ID: Mm00443258_m1 | |
| Commercial assay or kit | TaqMan mouse IL-10 gene expression assay | Thermo Fisher | Assay ID: Mm01288386_m1 | |
| Commercial assay or kit | TaqMan mouse HPRT gene expression assay | Thermo Fisher | Assay ID: Mm03024075_m1 | |
| Commercial assay or kit | BCA protein assay kit | Thermo Fisher | Cat# 23225 | |

*Continued on next page*

*Continued*

| Reagent type (species) or resource | Designation | Source or reference | Identifiers | Additional information |
|---|---|---|---|---|
| Commercial assay or kit | RC DC protein assay kit II | Bio-Rad | Cat# 5000122 | |
| Commercial assay or kit | OxyBlot Protein Oxidation Detection Kit | EMD Millipore | Cat# S7150 | |
| Chemical compound, drug | Imipenem; IPM | Primaxin IV; imipenem 500 mg stabilized in cilastatin | NDC 0006-3516-59 | 1.5 mg/mouse |
| Chemical compound, drug | Physiological Saline (0.9%) | Abbott Laboratories | Ref. # 04930-04-10 | |
| Software | ZEN (blue edition) imaging software | Zeiss | RRID:SCR_013672 | |
| Software | ImageJ software (version 1.46 r) | National Institutes of Health | RRID:SCR_003070 | |
| Software | Image Lab software (2017) | Bio-Rad | Cat# #1709690 | |
| Software | SAS 9.4 | SAS Institute Inc | | |
| Software | Aperio ImageScope software (12.4) | Leica | RRID:SCR_014311 | Positive-pixel algorithm |

## Animals and husbandry

Late-middle-aged adult C57BL/6 mice were acquired from the National Institute on Aging, and all experiments were initiated when animals were 16 months old (average male body weight ~34 grams, female body weight ~28 grams). Animals were acclimated in the Division of Laboratory Animal Resources at the University of Kentucky for at least 10 days before experimental procedures were performed. The mice were housed in pressurized intraventilated (PIV) cages with ad libitum access to drinking water and chow (Teklad Global No. 2918, Madison WI). Temperature (21–23°C), humidity (30–70%), and lighting (14/10 hr light/dark cycle) were controlled. All experimental procedures were approved by the Institutional Animal Care and Use Committee. All animal handling techniques were in accordance with the National Institutes of Health guidelines for ethical treatment.

## Chronic mouse model of sepsis

Experimental groups were allocated using restricted randomization to evenly distribute particularly large or small animals among control and sepsis groups. Profound polymicrobial sepsis was initiated by intraperitoneal (i.p.) injection of cecal slurry (CS; prepared as previously described in detail in *Starr and Saito, 2014*, with minor refinements as noted in *Steele et al. (2017)*, at a dose which was 100% lethal when administered without subsequent therapeutic resuscitation. Animals which did not develop severe hypothermia (≤30°C at 12h) were excluded from the study. Antibiotics (imipenem, IPM; 1.5 mg/mouse, i.p.) and fluid resuscitation (sterile physiological saline 0.9%) were administered beginning 12h following CS injection and continued twice daily. Antibiotic therapy was continued for at least 5 days, and fluid resuscitation (700 µL s.c.) was continued until body temperature recovered to ≥35.0°C.

Survival, body weight and body temperature were monitored regularly. Fat and lean mass were also monitored using EchoMRI Body Composition Analyzer (EchoMRI LLC, Houston, TX, USA) in a select group. Sepsis survivors were euthanized as previously described on days 4, 2 weeks (days 14–15), and 1 month (days 28–31) following CS injection along with non-sepsis controls as follows. Animals were anesthetized by inhalation of 5% isoflurane, and were maintained under 2.5% as a laparotomy was performed and blood was collected from the inferior vena cava (IVC) with a syringe containing 10% vol of 0.1M sodium citrate, and the IVC was cut to ensure exsanguination.

## Pair-feeding

Mice were singly housed and baseline food consumption was monitored daily at the same time of day for 5 days, and for 2 weeks after cecal slurry injection. Baseline food consumption was calculated as the average food consumed by all mice (*n* = 5 sepsis survivors) over the 5 day period, and daily averages were calculated after sepsis was induced. In a separate experiment, the food of non-sepsis animals was restricted according to the daily average food consumed by animals with sepsis (*i.e.* pair-fed mice); this experiment was conducted alongside freely-fed (ad libitum) controls (*n* = 5 per group).

## Assessment of bacteremia and inflammation

Bacteremia was regularly assessed by culture of small blood samples taken from the tail vein as previously described (*Steele et al., 2017*). In another experiment, systemic inflammation was assessed by measuring IL-6 in plasma samples obtained from animals sacrificed on day 4 and 2 weeks alongside non-sepsis controls using a high sensitivity ELISA kit (eBIOSCIENCE, Vienna, Austria). TNF$\alpha$ and IL-10 were assessed in non-sepsis controls and animals with sepsis on day 4 and 2 weeks (in the same mice) using Meso Scale Discovery (Rockville, Maryland) customized V-PLEX multiplex assay. These assays were conducted in singlet due to limited sample volumes.

## RNA isolation and quantitative RT-PCR

Frozen muscles (gastrocnemius) were homogenized with TRIzol reagent (Invitrogen, Carlsbad, CA), and total cellular RNA was purified using PureLink RNA Mini Kit (Life Technologies, Grand Island, NY). The concentration of the RNA was determined by measuring the absorbance at 260 nm using a NanoDrop ONE microvolume spectrophotometer (NanoDrop Technologies, Wilmington, DE), and integrity was confirmed through visualization of 18S and 28S RNA bands using an Agilent 2100 Bioanalyzer (Agilent Technologies, Santa Clara, CA). Equivalent amounts of RNA were reverse transcribed into complementary DNA using SuperScript III First-Strand Synthesis SuperMix (Life Technologies) according to the manufacturer's protocol. TaqMan assays were purchased from ThermoFisher Scientific and the quantitative reverse transcriptase-polymerase chain reaction was performed on a QuantStudio 3 (Applied Biosystems, Foster City, CA). Target gene expression was normalized to hypoxanthine-guanine phosphoribosyl transferase (HPRT) expression as an endogenous control, and fold change was calculated as 2-($\Delta\Delta$CT), using the mean $\Delta$CT of the Control group as a calibrator. TaqMan assays used were Mm00446190_m1, Mm00443258_m1, Mm01288386_m1, and Mm03024075_m1 for IL-6. TNF$\alpha$, IL-10, and HPRT, respectively.

## Muscle function analysis

The right hind limb was skinned and immediately placed in oxygenated (95% $O_2$-5% $CO_2$) Krebs-Ringer solution (118 mM NaCl, 4.4 mM KCl, 1.2 mM $MgSO_4$, 1.3 mM $NaH_2PO_4$, 2.5 mM $CaCl_2$, 25 mM $NaHCO_3$, and 10 mM D-Glucose; pH 7.4). The muscle bath was continuously oxygenated while the extensor digitorum longus (EDL) was dissected using a microscope, and tethers were placed on the proximal and distal tendons using braided silk suture (4-0). The muscle was freed from the leg, and mounted by attaching the tether at the distal end to a fixed hook and the proximal end to the lever arm of an ASI 300C-LR Aurora Scientific transducer system (Aurora, Ontario, Canada). The muscle was positioned between platinum electrodes and suspended in a temperature-controlled (25°C) chamber containing the Krebs-Ringer solution which was continually oxygenated, and allowed to acclimate for 5 min. Using an Aurora stimulator (model 701C), the muscle was subjected to electrical field stimulation, and resulting force output was recorded using ASI 610A Dynamic Muscle Control software. The optimum length (Lo) was found by adjusting the EDL length to maximum twitch force (1 Hz stimulation), which was measured using digital calipers. Maintaining the muscle at Lo, the force-frequency relationship was elucidated using stimulus frequencies of 1, 15, 30, 50, 80, 150, and 250 Hz (note that 250 Hz stimulations were most appropriate for maximum force production for the middle aged mice used in these studies, whereas 300 Hz stimulations are commonly used when using young animals). When the protocol was complete, the muscle was transferred from the apparatus to a muscle bath where the suture tethers where carefully removed, then the muscle was weighed. The tissue length and weight were then used to calculate the physiological cross-sectional area (*Brooks and Faulkner, 1988*) which was used to normalize force outputs (specific force).

## Immunohistochemistry and histochemistry

Upon euthanasia, hind-limb muscles were carefully dissected, embedded in a thin layer of optimal cutting temperature (OCT) medium, pinned at resting length to cork board, and immersed in liquid nitrogen-cooled isopentane. After freezing, the cork board was transferred to a bed of dry ice where the muscle was removed and quickly transferred to pre-chilled cryovials and then stored at −80°C until sectioning (8 µm).

### Fiber-type cross-sectional area analysis

Fiber-type-specific isoforms of myosin heavy chain (MyHC) were stained as previously described (*Fry et al., 2015*). The sections were incubated overnight at 4°C in primary antibodies acquired from Developmental Studies Hybridoma Bank (DSHB; Iowa City, IA, USA): MyHC type I (1:100; BA.D5; IgG2b), type IIa (1:100; SC.71; IgG1), and type IIb (1:100; BF.F3; IgM) (type IIx fibers remained unstained). Secondary antibody incubation included 1:250 anti-mouse IgG2b conjugated with Alexa Fluor 647 (#A21242), 1:500 anti-mouse IgG1 conjugated with Alexa Fluor 488 (#A21121), and 1:250 anti-mouse IgM conjugated with Alexa Fluor 555 (#A21426), all from Life Technologies (Carlsbad, CA, USA), for 1 hr at room temperature. Sections were post-fixed with methanol (5 min), and mounted with vector shield mounting media (Vector Labs, Burlingame, CA, USA). Entire muscle cross-sections were imaged using the tiles feature within Zeiss Zen Blue software interfaced with an AxioImager M1 upright fluorescent microscope (Göttingen, Germany) which is semi-automated. Fibers were assigned as type I, IIa, and IIb depending on intensity within the Cy5, FITC, or Texas Red channels, respectively, and unstained fibers were assumed as type IIx fibers. Cross-sectional area was analyzed on whole cross-sections (medial head of the gastrocnemius and soleus) using an interactive semi-automated analysis program using blue ZEN software (Zeiss, Göttingen, Germany). Representative images (20X magnification) are shown.

### Histochemical stains

The standard NADH (nicotinamide adenine dinucleotide) histochemical staining protocol was followed. Briefly, thawed skeletal muscle cross-sections were incubated in 2.4 mM NADH and nitro-blue tetrazolium (NBT) in 0.5 M Tris buffer for 30 min at 37°C. Tissues were fixed using 10% phosphate buffered formalin, washed with a series of acetone solutions, and cover-slipped using aqueous mounting medium (Vecta Mount AQ). Similarly, the SDH (succinate dehydrogenase) staining protocol was followed in which sections were incubated in 100 mM sodium succinate salt and 1.2 mM NBT in 0.2 M phosphate buffer for 1 hr at 37°C. The sections were fixed, rinsed, washed with acetone solutions, and cover-slipped as the NADH staining procedure. The COX (Cytochrome c oxidase) staining protocol was performed in which sections were incubated with cytochrome c (1.1 mM; Sigma C-2506), 342 mM sucrose, 4.3 mM catalase, and DAB (Dako) in 0.05 M phosphate buffer for 1 hr at 37°C. The sections were fixed and subsequently dehydrated and cover-slipped using Cytoseal 60 (Thermo Scientific) mounting medium.

Whole muscle cross-sections stained using NADH, SDH, and COX protocols were imaged in their entirety at 20X magnification using Aperio scanning software (Leica Biosystems Imaging Inc). These were quantified using positive-pixel algorithms using ImageScope software (version 11.2.0.780) for positive pixel intensity. Representative images were acquired using Nikon Eclipse E200 microscope outfitted with Nikon digital Sight DS-U3/DSFil software integrated with NIS Elements F3.2 Imaging Software.

## Transmission electron microscopy

The morphology of the mitochondria was evaluated by transmission electron microscopy as previously described (*Nakazawa et al., 2017*) with minor modifications. Briefly, extensor digitorum longus and tibialis anterior samples were collected and were immersed in fixation buffer comprised of 2.5% glutaraldehyde and 2.0% paraformaldehyde in 0.1 M sodium cacodylate buffer (pH 7.4) overnight. The tissue samples were post-fixed for 1 hr (1% osmiumtetroxide, 1.5% potassiumferrocyanide, stained for 1 hr (1% uranyl acetate), and dehydrated. Samples were infiltrated overnight in a 1:1 mixture of propylene oxide and TAAB Epon (Marivac Ltd., St. Laurent, Canada), and viewed and imaged under the Philips Technai BioTwin Spirit Electron Microscope (FEI, Hillsboro, OR, USA) at the Harvard Medical School Electron Microscopy facility (*Nakazawa et al., 2017*). At least five fields

of view of subsarcolemmal and intermyofibrillar mitochondria populations were captured for each sample using an AMT 2 k CCD camera in a blinded manner. ImageJ software (version 1.46 r, NIH) was used to perform morphometric analysis using the cell counter analysis plugin.

## Mitochondrial isolation and respiration analysis

Muscle mitochondria were isolated as described previously (*Patel et al., 2009*; *Gollihue et al., 2018*). Briefly, the tibialis anterior was quickly dissected and placed in ice cold isolation buffer (215 mM mannitol, 75 mM sucrose, 0.1% BSA, 20 mM HEPES, 1 mM EGTA; pH 7.2). The muscle was minced in trypsin (0.25 mg/ml) before homogenizing on ice in 3 rounds of 5 s intervals (total of 15 s; motor-driven Potter-Elvehjem homogenizer), and a protease inhibitor cocktail was added to the tissue homogenates. Mitochondrial pellets were obtained through differential centrifugation steps at 4°C. The pellet was suspended in isolation buffer and subjected to protein estimation was performed (BCA protein assay kit, Thermo Scientific). It was previously shown that this method yields pure mitochondria being that the resulting fraction does not express transaminase (*Patel et al., 2009*).

Mitochondrial respiration was assessed in terms of mitochondrial oxygen consumption rate (OCR) using Seahorse Bioscience XF$^e$24 extracellular flux analyzer, as previously reported (*Gollihue et al., 2018*; *Patel et al., 2014*; *Sauerbeck et al., 2011*). Briefly, the 24-well dual-analyzer sensor cartridges (Agilent Technologies, Santa Clara, CA, USA) were placed in a carbon dioxide-free incubator at 37°C the day prior to the experiment. Once the mitochondria were isolated the day of the experiment, the Seahorse Flux Pak cartridges were filled in the following manner: (A) pyruvate, malate, and ADP (to yield final concentrations of 5 mM, 2.5 mM, and 1 mM, respectively), (B) oligomycin (1 μg/mL), (C) carbonilcyanide p-triflouromethoxyphenylhydrazone (FCCP; 3 μM), and (D) rotenone plus succinate (100 nM and 10 mM, respectively). The Seahorse XF24 Flux Analyzer was calibrated as previously described (*Patel et al., 2014*) using mitochondrial protein and respiration buffer (125 mM potassium chloride, 2 mM magnesium chloride, 2.5 mM potassium phosphate monobasic, 20 mM HEPES, and 0.1% BSA, adjusted to pH 7.2). The experimental plates contained both non-sepsis control and post-sepsis samples in triplicate, and were subjected to OCR analysis through the series of dispensing solutions containing substrates and inhibitors that were added to injector ports A-D. Rates were generated using the AKOS oxygen consumption rate calibration algorithm; the average rate was determined for each sample well, and the replicates (three technical replicates per sample) were averaged.

## Protein extraction and immunoblotting

Upon euthanasia, skeletal muscles were dissected, placed in cryovials, snap frozen in liquid nitrogen, and stored at −80°C. Protein was extracted using the method described by *Feng et al. (2012)* with slight adaptations. Protein isolation buffer was comprised of 2% SDS, 10% glycerol, 2% 2-mercaptoethanol, in 50 mM Tris base, and a protease inhibitor cocktail was added (Sigma). The muscle samples were homogenized using dounce homogenizers in approximately 20 volumes (w/v) of isolation buffer. The homogenates were transferred to Eppendorf tubes, and heated (80°C water bath for five minutes), centrifuged (12,000 g for 10 min at room temperature), and the resulting supernatant was collected. The protein concentration of the samples was evaluated using the Bio-Rad *RC DC* assay (Hercules, CA, USA) according to the manufacturer's protocol.

The control and post-sepsis skeletal muscle protein isolates (20 μg loading protein) were resolved by SDS-PAGE electrophoresis (Bio-Rad Mini Protean Tetra Cell system) using TGX stain-free gradient (4–20%) gels, total protein was visualized using stain-free technology (ChemiDoc MP imaging system), and proteins were electrophoretically transferred (Bio-Rad Trans-blot Turbo Transfer System) to polyvinylidene difluoride (pvdf) membranes. Protein carbonyls were evaluated using the OxyBlot kit (EMD Millipore Corp, Billerica, MA, USA) by following the standard protocol with the minor adjustment of incubating in the primary antibody overnight (4°C). For 3-nitrotyrosine, the membranes were blocked for 1 hr in 5% milk at room temperature, and incubated in 1:3,000 3-nitrotyrosine primary antibody (Abcam #ab61392 in 5% milk) overnight at 4°C. Secondary antibody incubation (1:10,000 Santa Cruz #SC2005) was conducted for 1 hr at room temperature, and detected by chemiluminesce (Bio-Rad Clarity Western ECL Substrate). Densitometry analysis was performed on the resulting blots using Image Lab software (2017), and normalized to total protein analysis.

## Statistical analysis

Survival data were analyzed using Kaplan-Meier curves, with a Log Rank test to confirm significant differences between groups. In this case, and for all other data in this study, males and females were analyzed separately.

For outcome variables measured over time or with multiple observations taken from the same subject (such as specific force), a full-factorial repeated-measures ANOVA was performed, first analyzing overall differences across the various treatment groups. Likelihood ratio testing and Akaike Information Criterion (AIC) were used to select appropriate covariance structures in each case. When the interaction between treatment group and time was significant, relevant pairwise differences were calculated and reported with their respective $F$-statistics and $p$-values along with the full ANOVA table. A Kenward-Roger or Mancl-DeRouen adjustment was used, as appropriate, to correct for negative bias in the standard errors and degrees of freedom calculations induced by small samples.

The remaining outcomes of interest were measured for each subject only once. Thus, for these quantitative variables, we performed a one-way ANOVA model or unpaired $t$-test, as appropriate, to analyze differences in the response across the groups studied. For outcomes whose overall ANOVA $p$-value was significant (less than 0.05), relevant pairwise differences were calculated and reported with their respective $F$-statistics and $p$-values along with the full ANOVA table.

No outlier removal was performed, thus all data are presented. Sample sizes were estimated based on preliminary data comparing muscle strength of sepsis survivors and non-sepsis controls by ex vivo specific force analyses. With alpha = 0.05, and conservative estimation of standard deviation for treatment (i.e. sepsis) and control at 20, $n$ = 6 per group had 80% power to detect a difference in the group means. For mitochondrial respiration assessments, previous experiments determined that $n$ = 6 per group was sufficient to detect a 20% change from controls as statistically significant with 80% power (*Patel et al., 2014*). With these analyses, we aimed for groups sizes of at least $n$ = 6–7 for other experiments, anticipating consistent variation in the groups. This was not always achieved due to variation in sepsis survival rates. Due to feasibility, $n$ = 3 per group were used for transmission electron microscopy observation. Although underpowered, differences were still observed after post-hoc and small sample size corrections were performed, where appropriate, as detailed above.

Throughout the study, a $p$-value of less than 0.05 was considered significant. All analyses were completed in SAS 9.4 (SAS Institute Inc; Cary, NC, USA). Statistical analyses were performed by statisticians (AJS and GSH) in the University of Kentucky Department of Statistics.

## Acknowledgements

This project was mainly supported by NIH R01GM126181 (HS), R01 AG039732 (HS), R01 AG055359 (HS) and F31 GM117868 (AMO). The authors thank Ms Dana Napier and Karrie Jones with Biospecimen Procurement and Translational Pathology Shared Resource Facility of the University of Kentucky Markey Cancer Center (supported by the grant P30 CA177558) for their assistance in sectioning skeletal muscle specimens; Jennifer Moylan with the Center for Clinical and Translational Science (supported by the grant UL1TR001998) for assistance with multiplex assay analysis; members of the Center for Muscle Biology, especially, Dr. Sarah White for her time sharing knowledge on fiber-type staining and quantification. Body composition analysis was conducted using echoMRI technology supported by grant P20 GM103527. Aperio ScanScope imaging was conducted with the help of the Markey Cancer Center and Alzheimer's Disease Center supported by the grant P30 AG028383. Additionally, we thank Drs. Esther Dupont-Versteegden and Gerald Supinski for their helpful discussion in preparation for these experiments. Additionally, thanks to Ms. Donna Gilbreath of the Markey Cancer Center Research Communications Office for illustrative assistance.

## Additional information

### Funding

| Funder | Grant reference number | Author |
| --- | --- | --- |
| National Institute of General Medical Sciences | F31 GM117868 | Allison M Owen |
| National Institute of General Medical Sciences | R01 GM126181 | Hiroshi Saito |
| National Institute on Aging | R01 AG055359 | Hiroshi Saito |
| National Institute on Aging | R01 AG390732 | Hiroshi Saito |
| National Institute of General Medical Sciences | R01 GM129532 | Marlene E Starr |
| National Institute of General Medical Sciences | R01 GM117298 | Masao Kaneki |
| National Institute of General Medical Sciences | R01 GM115552 | Masao Kaneki |
| Shriners Hospitals for Children | 85600 | Masao Kaneki |

The funders had no role in study design, data collection and interpretation, or the decision to submit the work for publication.

### Author contributions

Allison M Owen, Conceptualization, Data curation, Formal analysis, Investigation, Visualization, Writing—original draft, Writing—review and editing; Samir P Patel, Marlene E Starr, Conceptualization, Resources, Data curation, Supervision, Funding acquisition, Investigation, Visualization, Methodology, Writing—original draft, Project administration, Writing—review and editing; Jeffrey D Smith, Conceptualization, Resources, Data curation, Supervision, Funding acquisition, Investigation, Methodology, Writing—original draft, Project administration, Writing—review and editing; Beverly K Balasuriya, Conceptualization, Data curation, Investigation, Methodology, Writing—review and editing; Stephanie F Mori, Data curation, Investigation, Writing—review and editing; Gregory S Hawk, Data curation, Investigation; Arnold J Stromberg, Data curation, Formal analysis, Investigation, Writing—original draft, Writing—review and editing; Naohide Kuriyama, Resources, Data curation, Formal analysis, Supervision, Writing—original draft, Writing—review and editing; Masao Kaneki, Resources, Software, Formal analysis, Supervision, Investigation, Methodology, Writing—review and editing; Alexander G Rabchevsky, Resources, Software, Supervision, Funding acquisition, Investigation, Methodology, Writing—review and editing; Timothy A Butterfield, Conceptualization, Resources, Supervision, Funding acquisition, Writing—review and editing; Karyn A Esser, Charlotte A Peterson, Conceptualization, Resources, Software, Supervision, Writing—review and editing; Hiroshi Saito, Conceptualization, Resources, Supervision, Funding acquisition, Investigation, Writing—original draft, Project administration, Writing—review and editing

### Author ORCIDs

Allison M Owen (iD) https://orcid.org/0000-0001-5623-4891
Karyn A Esser (iD) http://orcid.org/0000-0002-5791-1441
Marlene E Starr (iD) https://orcid.org/0000-0003-0304-348X
Hiroshi Saito (iD) https://orcid.org/0000-0002-5845-5177

### Ethics

Animal experimentation: All animal experiments presented in this study were conducted according to procedures outlined in the approved Institutional Animal Care and Use Committee (IACUC) protocol #2009-0541 of the University of Kentucky.

Decision letter and Author response
Decision letter https://doi.org/10.7554/eLife.49920.023
Author response https://doi.org/10.7554/eLife.49920.024

# Additional files

## Supplementary files
• Transparent reporting form DOI: https://doi.org/10.7554/eLife.49920.021

## Data availability
No datasets were generated in preparation of this manuscript.

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
