## [Decision Letter]

**Acceptance summary:**

We believe your paper represents a significant advance in understanding the longer term effects of sepsis on muscle weakness and points to changes in muscle mitochondrial structure and function as a residual effect of a major acute insult. The previous challenge in modeling mouse models for chronic sepsis has been the issue of long term survival. Your cecal slurry and antibiotics model insures longer term survival of mice and allows a clearer delineation of muscle structure after muscle mass restoration. In your paper, your group clearly demonstrate altered mitochondrial structure, electron chain dysfunction, impaired respiration and oxidative damage in muscle despite restoration of muscle mass. Importantly, you are able to demonstrate these changes are not nutritionally mediated with paired feeding. As such these data have translational implications for understanding muscle weakness, and long term morbidity, after serious acute ICU-illnesses. This paper should open the door to more comprehensive studies for our understanding of mitochondrial dysfunction after major stress. As such there are important translational implications.

**Decision letter after peer review:**

Thank you for submitting your article "Chronic muscle weakness and mitochondrial dysfunction in the absence of sustained atrophy in a preclinical sepsis model" for consideration by *eLife*. Your article has been reviewed by two peer reviewers, and the evaluation has been overseen by Clifford Rosen as the Senior and Reviewing Editor. The following individual involved in review of your submission has agreed to reveal their identity: Edward Sherwood (Reviewer #1).

The reviewers have discussed the reviews with one another and the Reviewing Editor has drafted this decision to help you prepare a revised submission.

Summary:

Overall there is general agreement that the sepsis model developed in your manuscript provides new insights into sepsis related mitochondrial dysfunction. The experiments are well done and the conclusions appropriate. However there are several concerns which limit enthusiasm and must be addressed.

Essential revisions:

1) Food intake was not reported among the mice, yet might be a critical factor in the changes related to mitochondrial dysfunction.

2) The underlying mechanism(s) are not clarified and an unbiased approach using RNAseq or proteomics might provide clues and we would request more information.

3) The mitochondrial bioenergetics are lacking some critical data – what about glycolytic rates? Were flux studies performed? These need to be included.

Reviewer #1:

The paper by Owen and colleagues examines muscle function and energetics in septic mice. The authors created a clinically relevant model of sepsis survival using intraperitoneal cecal slurry injection and resuscitation with fluids and antibiotics. The authors report that surviving mice had significant weakness and muscle dysfunction associated with impaired mitochondrial function. This is a well done experimental study that addresses a clinically important problem and provides mechanistic insight.

Overall, this is a nicely done study that provides evidence of muscle mitochondrial dysfunction in sepsis survivors. The paper left me wanting more information about glucose uptake, glycolytic rate and TCA cycle flux. It also leaves me wondering about the mechanisms of mitochondrial injury and the level of mitophagy. Nevertheless, this is a well done preliminary study that the authors will hopefully build upon in future work.

Reviewer #2:

The manuscript by Owen et al. addresses a clinically significant problem of reduced function and high morbidity in patients who survive sepsis. This problem is understudied relative to the disease burden. This manuscript is an excellent, foundational study that documents the muscle phenotype in a mouse model of sepsis. The strengths of the study include the use of a reproducible model of sepsis survival, the use of appropriate controls, the use of adult mice (16 months), the characterization of both males and females, and the use of gold standard analytical procedures to measure muscle endpoints. The manuscript is well-written, the methods are well-described, and the authors are to be commended on an important contribution to the literature.

In their investigation, the authors find reduction of muscle force, but not muscle mass at 2 and 4 weeks after the initiation of sepsis. They demonstrate reduced mitochondrial respiration, ultra-structural defects, reduced mitochondrial enzymatic activity, and evidence of oxidative damage. This suggests the proximal cause of muscle weakness is mitochondrial dysfunction. All of this is a reasonable explanation for the sustained muscle dysfunction. There are no interventions or specific molecular mechanisms that are tested and implicated in this process, although sustained infection/inflammation is ruled out.

Some concerns and missed opportunities that could be addressed in the Discussion:

1) Reproducibility of the model in others' laboratories. Given that the microbiome can influence multiple endpoints and that this can be animal-facility specific, one wonders if a distant lab would be able to find the same survival curves and subsequent effects. Some discussion of this would be welcome.

2) While the model is a welcome addition to the literature and displays strengths as enumerated, the authors do not describe a plausible clinical scenario that it is intended to mimic, limiting significance.

3) The model does not reproduce the muscle wasting observed in post-sepsis/ICU patients. This might be useful experimentally but is a limitation.

4) The use of the term "sepsis survivors" in the Discussion elicits thoughts of patients. However, there are no human samples in the study; thus, the mechanism is still not generalizable to the experience of patients. This limitation should be addressed.

5) There are no data on pair feeding or food intake in these mice. It is formally possible that the effects on muscle are due to a period of starvation. This limitation should be addressed experimentally, if possible.

This is a suggestion and a hope:

6) An -omics approach of any sort – shotgun or targeted proteomics, RNAseq, etc. – could point to possible specific mechanisms dysregulating the mitochondria, would strengthen the impact and, if reported and deposited to GEO, would be a substantial contribution to the field. Such a contribution likely would enable others to query pathways of interest and thus would spark additional research using this model specifically and also more generally in the arena of sepsis-related muscle weakness/wasting.

---

## [Author Response]

Essential revisions:1) Food intake was not reported among the mice, yet might be a critical factor in the changes related to mitochondrial dysfunction.2) The underlying mechanism(s) are not clarified and an unbiased approach using RNAseq or proteomics might provide clues and we would request more information.3) The mitochondrial bioenergetics are lacking some critical data – what about glycolytic rates? Were flux studies performed? These need to be included.Reviewer #1:[…] Overall, this is a nicely done study that provides evidence of muscle mitochondrial dysfunction in sepsis survivors. The paper left me wanting more information about glucose uptake, glycolytic rate and TCA cycle flux.

We thank the reviewer for raising this important question regarding glycolysis and flux measurements, and agree that it is important to further delineate alterations in mitochondrial metabolism in the post-sepsis state. Although it is important to investigate, these measurements require whole cells. A limitation of our study, we conducted seahorse measurements on isolated mitochondria from muscle tissue since isolation and use of whole muscle cells (i.e. myofibers) introduce several technical concerns (length of myofibers relative to the diameter of the assay well, obtaining a single cell suspension of cells of varying myofiber length, normalization challenges etc.). Using isolated mitochondria, we were not able to measure the extracellular acidification rate (ECAR), as the reviewer is pointing towards. Important to note, we understand that assessing oxidative phosphorylation in isolated mitochondria does not fully reflect the physiological situation due to being outside of the natural cellular environment; thus, we also assessed mitochondrial enzyme activities in whole tissue using histochemical stains. With respect to assessments of glycolysis in the post-sepsis, we plan to conduct a metabolomics study and assess flux in tissue after feeding C13 labeled glucose. However due to the laborious nature of the experiments and large amounts of data to be produced, we were not able to include these data for the resubmission within the given period of time. We have included comments as such in the Discussion (sixth paragraph), and look forward to making this contribution to the field in the near future.

It also leaves me wondering about the mechanisms of mitochondrial injury and the level of mitophagy.

Likewise, we are interested in exploring the mechanisms of mitochondrial injury and investigating mitochondrial dynamics in terms of mitophagy and turnover. Although we have begun pilot experiments concerning these matters, these studies are rather complex due to (1) the multi-phasic nature of mitochondrial injury during the acute and chronic phases of sepsis pathogenesis, and (2) the added complication of using aged animals, which have impaired mitochondrial turnover. Thus, these issues will take a significant amount of time and resources to explore accurately. We are currently working on these experiments and aim to report on the mechanisms of mitochondrial injury and mitophagy as we build on the current work, to which we remark upon in the Discussion (sixth paragraph).

Reviewer #2:[…] Some concerns and missed opportunities that could be addressed in the Discussion:1) Reproducibility of the model in others' laboratories. Given that the microbiome can influence multiple endpoints and that this can be animal-facility specific, one wonders if a distant lab would be able to find the same survival curves and subsequent effects. Some discussion of this would be welcome.

This is an important point, and we thank the reviewer for giving us the opportunity to remark upon the reproducibility of our animal model of sepsis with ICU-like resuscitation. We would first like to point out that the issue of reproducibility is not unique to our sepsis model, as it is likewise a concern for the commonly used CLP surgical model. In fact, one of the strong advantages of the cecal slurry model of sepsis is the reduction of the potential influence of microbiomes in the cases where different kinds of mice are studied (i.e. transgenic mice vs. wild type, different ages, different diets, or disease states) since one batch of cecal slurry is prepared and used among all experimental animals. However, to address the reviewer’s important point more specifically: we are currently aware of at least two labs at different institutions who have successfully made their own cecal slurry stocks, conducted dose response experiments, and compared survival between animals with and without intervention. One research laboratory at the University of California San Francisco achieved a strikingly similar ~75% survival rate with the therapeutic intervention strategy described. Further, they confirmed bacteremia in all mice at 12-hours (before therapeutic intervention), and similar rates of bacterial clearance during the resuscitation period. Likewise, a group at Vanderbilt University Medical Center adopted the model which they adapted to earlier therapeutic intervention windows (3- and 6h) to address their research question, which resulted in 100% and 80% survival, respectively. These findings are encouraging that this model is reproducible in other hands, and are hopeful that many labs will be able to adopt our model to address their research questions.

2) While the model is a welcome addition to the literature and displays strengths as enumerated, the authors do not describe a plausible clinical scenario that it is intended to mimic, limiting significance.

We thank the reviewer for their kind remarks regarding our model, and apologize for not providing clarity regarding the clinical scenario that it is intended to mimic. We provide further explanation to this end in the revision, which can be found in the third paragraph of the Discussion.

3) The model does not reproduce the muscle wasting observed in post-sepsis/ICU patients. This might be useful experimentally but is a limitation.

Indeed, our model does not reproduce *long-term* muscle wasting as observed commonly in post-sepsis/ICU patients. It is first important to highlight that not all muscle weakness after hospital discharge is accompanied by muscle wasting, and these are separate issues that we suggest be considered independently. However, we recognize that atrophy is a common issue and primary concern in the post-ICU condition. We presume that a key reason this phenomenon is not reproduced in our model is the robustness of murine subjects, which become highly active very quickly even after serious illness, thus the state of bedrest is not incorporated in our model. Also, many sepsis patients are mechanically ventilated, preventing physical activity even further. It is our interest to expand our model further in the future to incorporate experimental procedures to further mimic the clinical setting, such as corticosteroid treatment, immobilization, and mechanical ventilation, and determine if this replicates the muscle wasting phenotype observed commonly in sepsis survivors. We have added remarks on this matter to the Discussion (fourth paragraph).

4) The use of the term "sepsis survivors" in the Discussion elicits thoughts of patients. However, there are no human samples in the study; thus, the mechanism is still not generalizable to the experience of patients. This limitation should be addressed.

We apologize for using the term “sepsis survivors” in the Discussion, implying the use of patient samples. Thank you for bringing this to our attention, we have edited the text to more accurately reflect the scope of our study.

5) There are no data on pair feeding or food intake in these mice. It is formally possible that the effects on muscle are due to a period of starvation. This limitation should be addressed experimentally, if possible.

We conducted an experiment to address this logical concern. First, we monitored daily food consumption of middle-aged mice (*n*=9) before and after inducing sepsis via our cecal slurry and resuscitation model. Five of the nine mice survived for 14-days, from which we averaged the daily food intake. Food consumption was dramatically reduced on the day of infection (11.08% of pre-sepsis food intake), which steadily increased to 75.51% by day 4, and returned to comparable consumption thereafter. Next, we restricted food intake of middle-aged non sepsis animals (n=5) to match the daily food consumption by the sepsis animals. Freely-fed non sepsis mice served as controls during the same period (*n*=5). Non-sepsis pair-fed mice unsurprisingly lost substantial body weight. After 14-days of pair-feeding, skeletal muscle tissue (*tibialis anterior*) was harvested, mitochondria was isolated, and mitochondrial respiration capacity was evaluated. The results show that oxygen consumption rate was similar among pair-fed and freely-fed control mice (new figure added: Figure 5—figure supplement 1). From these data, we confirmed that reduced food intake and weight loss during sepsis is not the underlying cause of long-term mitochondrial dysfunction in the murine sepsis survivors at this late time point (2 weeks).

This is a suggestion and a hope:6) An -omics approach of any sort – shotgun or targeted proteomics, RNAseq, etc. – could point to possible specific mechanisms dysregulating the mitochondria, would strengthen the impact and, if reported and deposited to GEO, would be a substantial contribution to the field. Such a contribution likely would enable others to query pathways of interest and thus would spark additional research using this model specifically and also more generally in the arena of sepsis-related muscle weakness/wasting.

We agree with the reviewer that an -omics approach would provide many insights to the innumerable pathways which are sure to be affected in the murine sepsis survivors. However, with limited aged mice available to our lab during this revision period, and with two experiments required to address the above concern in point #1, we could not perform this set of studies this time. We have included this point in our future direction plan, which we remark upon in the Discussion (fourth paragraph).